# Mycobacterial DnaB helicase intein as oxidative stress sensor

Danielle S. Kelley[1], Christopher W. Lennon[2], Zhong Li[3], Michael R. Miller[4], Nilesh K. Banavali[1,3], Hongmin Li ⓘ [1,3] & Marlene Belfort[1,2]

Inteins are widespread self-splicing protein elements emerging as potential post-translational environmental sensors. Here, we describe two inteins within one protein, the *Mycobacterium smegmatis* replicative helicase DnaB. These inteins, DnaBi1 and DnaBi2, have homology to inteins in pathogens, splice with vastly varied rates, and are differentially responsive to environmental stressors. Whereas DnaBi1 splicing is reversibly inhibited by oxidative and nitrosative insults, DnaBi2 is not. Using a reporter that measures splicing in a native intein-containing organism and western blotting, we show that $H_2O_2$ inhibits DnaBi1 splicing in *M. smegmatis*. Intriguingly, upon oxidation, the catalytic cysteine of DnaBi1 forms an intramolecular disulfide bond. We report a crystal structure of the class 3 DnaBi1 intein at 1.95 Å, supporting our findings and providing insight into this splicing mechanism. We propose that this cysteine toggle allows DnaBi1 to sense stress, pausing replication to maintain genome integrity, and then allowing splicing immediately when permissive conditions return.

[1] Department of Biomedical Sciences, School of Public Health, University at Albany, Albany, NY 12222, USA. [2] Department of Biological Sciences and RNA Institute, University at Albany, Albany, NY 12222, USA. [3] Wadsworth Center, New York State Department of Health, 120 New Scotland Ave, Albany, NY 12208, USA. [4] Department of Chemistry, University at Albany, Albany, NY 12222, USA. These authors contributed equally: Danielle S. Kelley, Christopher W. Lennon. Correspondence and requests for materials should be addressed to H.L. (email: hongmin.li@health.ny.gov) or to M.B. (email: mbelfort@albany.edu)

nteins are dynamic intervening protein elements that invade at the DNA level and are transcribed and translated along with the host protein. Long thought to be strictly parasitic, recent work has challenged this notion and suggested that inteins can function as post-translational sensors that respond to environmental cues[1–6]. Inteins have the unique ability to catalyze their own excision from the host protein, ligating the two flanking peptide sequences, termed exteins, together to form the mature protein[7]. In addition to splicing, inteins often contain a homing endonuclease domain (HEN) which allows inteins to spread[8]. As such, these mobile elements have been found in all three domains of life across a wide-array of microbial genomes[9–11], particularly abundant among bacteria and archaea in essential genes, as well as in viruses and phages[12–14].

Of interest are inteins in mycobacteria, which have been shown to be a highly intein-rich genus[9,10]. Mycobacteria variably contain six different intein-containing proteins, and these proteins perform many critical functions in the cell, including roles in DNA replication and repair, iron–sulfur cluster biogenesis, and stress response[9,10,15,16]. Important pathogens, such as *Mycobacterium tuberculosis* and *Mycobacterium leprae*, and non-pathogenic models, like *Mycobacterium smegmatis*, all contain multiple inteins, although the intein distribution is species specific. Even inteins found in the same protein can differ in position and sequence between host bacteria and, in some instances, multiple inteins are inserted into a single gene, such as the *dnaB* gene in *M. smegmatis*[9,10].

DnaB is the replicative helicase in bacteria and an essential component of the replication machinery. Additionally, DnaB is one of the most common intein-containing proteins among bacteria[10]. This helicase unwinds DNA in the 5′ to 3′ direction at the replication fork, playing a crucial role in replication initiation[17–19]. DnaB is composed of two domains: an N-terminal globule involved in protein–protein interactions that allow formation of a hexameric ring and a C-terminal ATPase responsible for the DNA-unwinding[18,20,21]. Inteins are prevalent among ATPase domains across distinct proteins and this is true for DnaB, which has intein insertions in the ATPase domain at three distinct sites (a, b, and c)[9,10]. In mycobacteria, inteins are found in two of the three sites (a and b)[9]. Inteins, which range in size from ~130 to over 800 amino acids[9,11], are considered disruptive to protein activity and prior to intein splicing the host protein is assumed to be functionally compromised. Understanding the biological impact of inteins on the host protein and organism is imperative to addressing the larger question of why inteins have been consistently maintained in certain locations across different organisms.

Intein maintenance has been attributed to the difficulties associated with precisely removing the intein without fatally inactivating the host protein and the stability of the insertion site sequences, as inteins are often found in highly conserved regions of proteins[8]. However, there is mounting evidence that inteins are not just selfish mobile parasites but can serve a post-translational regulatory role under specific conditions, potentially contributing to long-term persistence. Inteins have been found to be responsive to a range of stressors and environmental conditions, including temperature[4], DNA damage[2], salt[5,22], redox[6,23], and reactive oxygen and nitrogen species (ROS/RNS)[1,3]. These conditions are often highly relevant to the environmental niche of the organism, such as salt with a halophile[5,22], or they relate to the function of the intein-containing protein, like a RadA recombinase intein and its enhanced splicing in the presence of the RadA substrate ssDNA[2]. ROS and RNS stressors are of interest due to their relevance to the lifestyle of many mycobacterial species. Pathogens, such as *M. tuberculosis* and *M. leprae*, face these stressors during infection after exposure to the respiratory burst

by host macrophages[24–26]. Furthermore, a recent study showed that the intein in iron–sulfur scaffold protein SufB of *M. tuberculosis* is highly sensitive to splicing inhibition by oxidation and modifications caused by ROS and RNS stressors[3].

Here, we focus on the two inteins present in the *M. smegmatis* DnaB protein to address the potential for conditional splicing. Dramatic differences are found between the two inteins, DnaBi1 and DnaBi2, with respect to both their splicing rate and response to stressors. The mechanism of inhibition for DnaBi1 with ROS is elucidated, revealing that the catalytic cysteine engages in disulfide bond formation with a non-catalytic cysteine. We find that DnaBi1 splicing is inhibited under $H_2O_2$ stress in vivo using a reporter system in *M. smegmatis*, providing a measure of splicing in the native intein-containing host. Further, $H_2O_2$-based splicing inhibition is detected by western blotting in *M. smegmatis*. The crystal structure of class 3 DnaBi1 is solved to a resolution of 1.95 Å and shows that the catalytic cysteine adopts two different conformations. While the two cysteines are separated in the structure, only minor conformational changes are required for disulfide bond formation. These results highlight that inteins present within the same protein can behave distinctly, and that a redox-sensitive catalytic residue acts as a sensor to orchestrate conditional intein splicing. Thereby, intein regulation likely safeguards against replication stress when ROS are prevalent.

## Results

**The two DnaB inteins in *M. smegmatis* are distinctive.** The two inteins in the *dnaB* gene of *M. smegmatis* (*Msm*) are different from each other in sequence, insertion site, and splicing mechanism (Fig. 1). The first *Msm* intein, DnaBi1, lacks a HEN, necessary for invasion of novel sites, and is considered a mini-intein (Fig. 1a). This intein localizes to the P-loop of the DnaB ATPase domain at insertion site b, where the P-loop serine that participates in $Mg^{2+}$ coordination in the mature protein also serves as a catalytic residue for intein splicing (Fig. 1b, c). The second *Msm* intein, DnaBi2, contains a HEN for mobility (Fig. 1a), and is found at insertion site a in motif H4, a DNA-binding loop (Fig. 1b)[10,27]. The *Msm* inteins have homology to single inteins in pathogens *Mycobacterium leprae* (*Mle*) and *Mycobacterium tuberculosis* (*Mtu*). The DnaBi1 and *Mle* inteins share 68.0% amino acid identity and the DnaBi2 and *Mtu* inteins have 61.0% amino acid identity (Fig. 1a). These inteins share many defining features across species, including insertion site location, absence or presence of a HEN, and splicing mechanism, described below.

A major difference between the two DnaB inteins is the mechanism of splicing. DnaBi2 and its *Mtu* homolog splice by the canonical class 1 mechanism (Fig. 1c, bottom; Supplementary Fig. 1a). Class 1 inteins use a conserved nucleophile, cysteine, or serine, at the start of the intein sequence to initiate splicing. For *Msm* DnaBi2 and *Mtu* DnaBi this residue is a cysteine (Cys1) (Fig. 1c, bottom), which nucleophilically attacks the preceding amide bond at the N-extein–intein junction. A labile thioester linkage between the intein and the N-extein forms (step 1). The labile bond then undergoes a second nucleophilic attack by the first residue of the C-extein, in this case a serine (Ser + 1) (Fig. 1c, bottom, step 2). This transfers the N-extein to the C-extein, forming a branched intermediate. The branched intermediate resolves, forming a native peptide bond between the two exteins (steps 3 and 4) (Supplementary Fig. 1a).

*Msm* DnaBi1 and its *Mle* homolog splice by the class 3 pathway. The splicing pathway is coordinated by a set of conserved residues found in four blocks in all inteins (A, B, F, and G) that make up the splicing domain. Class 3 inteins lack a nucleophilic residue at the start of the intein sequence, instead

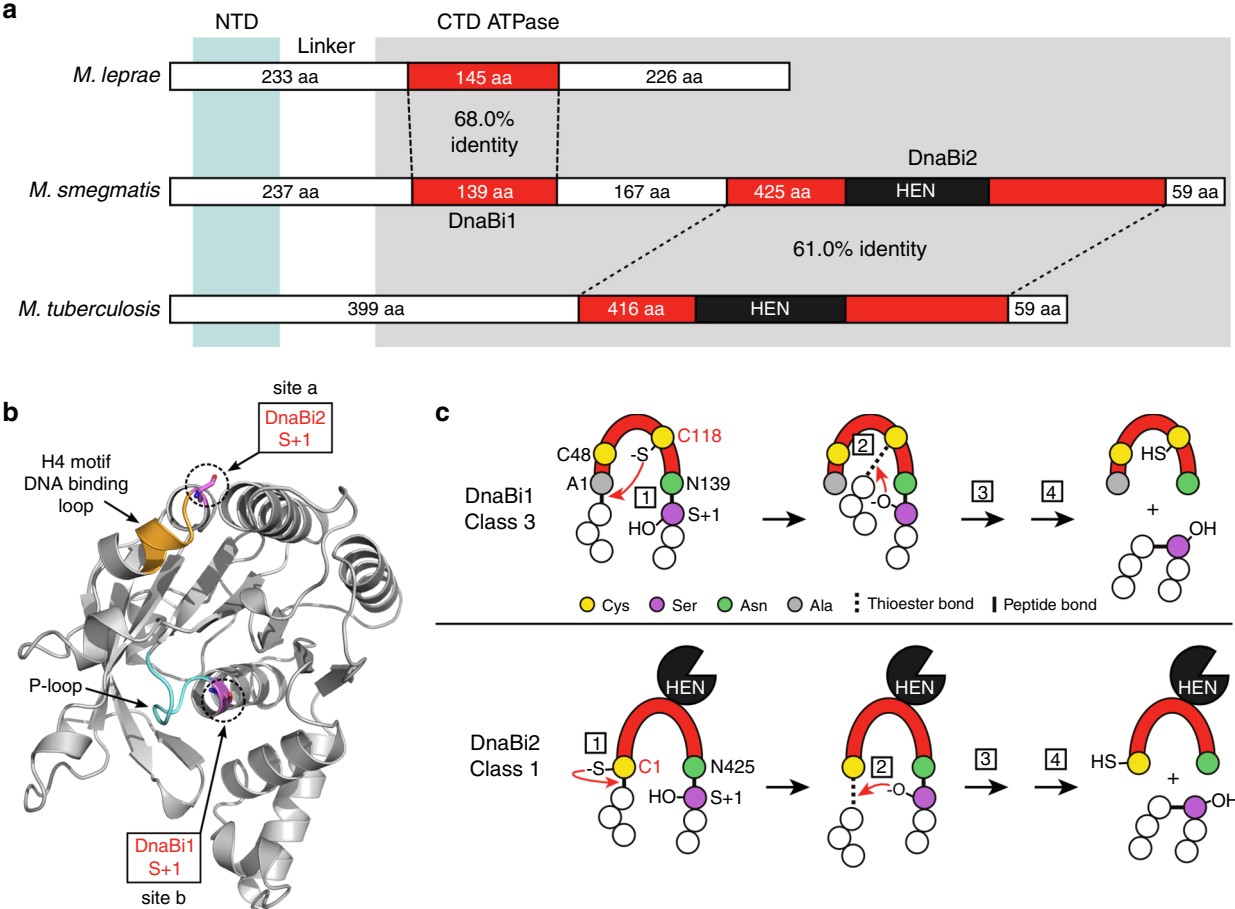

**Fig. 1** Overview of mycobacterial DnaB inteins. **a** Relationship of DnaB inteins in three mycobacterial species. The two *M. smegmatis* (*Msm*) DnaB inteins (DnaBi1 and DnaBi2) were aligned to the single DnaB inteins in *M. leprae* (*Mle*) and *M. tuberculosis* (*Mtu*) using a protein pairwise alignment (EMBOSS Needle; http://www.ebi.ac.uk/Tools/psa/emboss_needle). The amino acid percent identity between inteins (red) are shown. The number of residues for the intein and extein fragments (white) are indicated and the homing endonuclease (HEN) is shown in black. The N-terminal domain of DnaB is indicated in light blue shading and the C-terminal ATPase is indicated in gray shading. NTD N-terminal domain, CTD C-terminal domain, aa amino acid. **b** Localization of DnaB intein insertions. A structure model of the DnaB ATPase domain (residues 200–461) is shown[54]. The insertion sites of DnaBi1 and DnaBi2 are indicated by the S + 1 in purple. The P-loop, where DnaBi1 is inserted, is cyan and the H4 motif/DNA-binding loop, where DnaBi2 is found, is orange. **c** Intein splicing mechanisms. The class 3 mechanism (top) is used by *Msm* DnaBi1 and *Mle* DnaBi, whereas the canonical class 1 pathway (bottom) is used by *Msm* DnaBi2 and *Mtu* DnaBi. See the main text and Supplementary Figure 1 for detailed splicing description and steps (boxed numbers). Residue numbering refers to *Msm* inteins

using a conserved internal block F cysteine (Fig. 1c, top). This internal cysteine (Cys118 for DnaBi1) attacks the N-extein–intein junction (step 1), akin to Cys1 of class 1 inteins. This results in a branched intermediate lacking at this stage in the class 1 pathway (Fig. 1c). A second nucleophilic attack by the +1 serine (Ser + 1) occurs (step 2) and the pathway proceeds in a manner similar to class 1 (steps 3 and 4; Supplementary Fig. 1b), resulting in excised intein and ligated exteins (Fig. 1c, top).

***M. smegmatis* DnaB inteins have different splicing profiles**. To understand the splicing behavior of the two *Msm* DnaB inteins, DnaBi1 and DnaBi2 were cloned into splicing reporter MIG (maltose-binding protein-intein-GFP)[3]. MIG uses in-gel fluorescence to monitor splicing, allowing visualization of all GFP-containing products (Fig. 2a). Cell pellets with induced MIG reporter were lysed, representing time 0, and splicing was monitored over time. The two inteins have strikingly dissimilar splicing profiles (Fig. 2b). MIG DnaBi1 splices slowly and even after 24 h the splicing reaction has only gone to ~50% completion, with no major off-pathway cleavage products (Fig. 2b). In contrast,

MIG DnaBi2 splices rapidly, with the reaction having gone to completion by time 0, when cells are harvested (Fig. 2b). These results are mirrored by the DnaB inteins from *Mle* and *Mtu* (Supplementary Fig. 2).

This difference in splicing could be attributed to the foreign extein sequences used in MIG. While these fusion proteins contain 7 to 10 native flanking extein residues, long-range extein effects have been shown to influence intein splicing[4]. Therefore, we made a construct expressing wild-type (WT) full-length DnaB protein with the two splicing-active inteins. A mutant version with splicing-inactivating mutations in both inteins (DnaBi1, C118A/N139A; DnaBi2, C1A, N425A) was made, representing the full precursor polypeptide ($P_{i1,i2}$). Two alternative precursor products that could arise from either intein splicing were also generated, with splicing-inactivating mutations. These are alternative precursor 1 ($P_{i1}$), where intein 1 is present but not intein 2, and alternative precursor 2 ($P_{i2}$), where intein 1 is absent and intein 2 is present. Finally, a ligated extein (LE) construct lacking both inteins was made. The full-length WT DnaB with splicing-competent inteins accumulates primarily as $P_{i1}$, corresponding to DnaBi1 present and DnaBi2 having spliced out (Fig. 2c). The

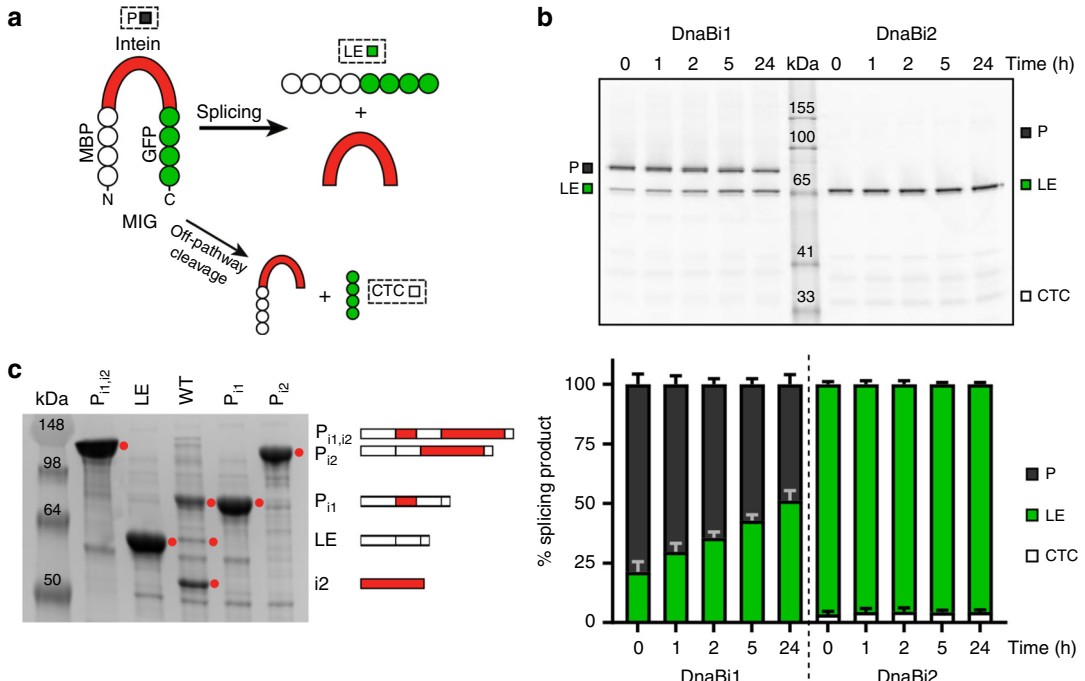

**Fig. 2** Different splicing profiles of *M. smegmatis* DnaB inteins. **a** Schematic of MIG. The reporter construct maltose-binding protein (MBP)-intein-GFP (MIG) allows monitoring of splicing by in-gel fluorescence of GFP-containing products[3]. The precursor molecule (P) can splice, yielding ligated exteins (LE) and free intein (I), or can undergo off-pathway cleavage reactions, such as C-terminal cleavage (CTC). **b** The two *Msm* inteins have distinct splicing. The gel of a splicing time course shows that MIG DnaBi1 splices slowly while MIG DnaBi2 splices almost instantaneously. Quantitation of MIG time course is shown below (stack plots), where the ratios of splice products were plotted. Data are representative of three biological replicates and values are expressed as mean ± s.d. **c** Splicing of *Msm* DnaB inteins with native exteins corresponds to splicing in MIG. Full-length DnaB protein constructs were made to understand how the inteins splice with native exteins. The wild-type (WT) lane represents lysate of overexpressed DnaB protein with splicing-competent inteins. The adjacent lanes are lysates containing splicing-inactive controls representing possible splicing outcomes and are schematically indicated. These splicing products include full precursor with both inteins present ($P_{i1,i2}$), alternative precursor 1 ($P_{i1}$), with only DnaBi1 present, alternative precursor 2 ($P_{i2}$), with only DnaBi2 present, and ligated exteins (LE), with no inteins present. In WT there is accumulation of $P_{i1}$, which indicates rapid splicing of DnaBi2. Consistent with this observation, abundant DnaBi2 is visible in the WT lane. Bands of interest are indicated by red circles. Data are representative of three biological replicates

large amount of $P_{i1}$ in WT suggests DnaBi1 splicing is slow while DnaBi2 splicing is fast in a native extein context. This mirrors how the individual inteins behaved in MIG (Fig. 2b) and indicates that the difference in splicing is likely intrinsic to the two *Msm* DnaB inteins, not a result of non-native exteins.

**Two inteins display differential sensitivity to stressors.** Recent work has shown that an *Mtu* intein is sensitive to inhibition by ROS and RNS due to modifications on the catalytic cysteine residues[3]. We therefore asked if the *Msm* inteins, which both utilize cysteine as the initiating nucleophile but not as the secondary nucleophile (Fig. 1c), may be similarly sensitive. Prior work has shown that changing the last residue of the N-extein can alter splicing kinetics[28]. Since MIG DnaBi2 splices rapidly, a random mutagenic screen of this N-extein residue, Gly-1, was performed to find alternative amino acids that reduced splicing rate. Mutant G-1V was isolated, which slows splicing enough to allow visualization of precursor and observe the difference in product ratios while not accumulating excessive off-pathway cleavage product. MIG cell lysate was treated with two ROS-generating stressors, $H_2O_2$ and diamide (DA), and two RNS-generating compounds, DEA NONOate (DEA) and Angeli's Salt (AS).

We again observed differences between the two DnaB inteins. MIG DnaBi1 was generally sensitive to splicing inhibition by ROS and RNS stressors while MIG DnaBi2 G-1V was not (Fig. 3). Treatment of MIG DnaBi1 with either ROS reagent $H_2O_2$ or DA

caused the appearance of a secondary product above the precursor band, while RNS reagents DEA and AS resulted in the precursor band appearing diffuse compared to the untreated control (Fig. 3a, top). Except for 0.8 mM $H_2O_2$, the treatments resulted in a substantial increase in the amount of precursor relative to the untreated sample (Fig. 3a, bottom) and no off-pathway cleavage was observed. In contrast, MIG DnaBi2 G-1V appeared largely unresponsive to inhibition by these stressors even at a higher magnification or increased contrast, but we cannot exclude the possibility that splicing is occurring too rapidly for these compounds to show an effect.

The appearance of a higher, secondary precursor and band diffuseness with MIG DnaBi1 could be indicative of reversible cysteine modifications, such as disulfide bond formation or nitrosylation. To determine if the observed changes were due to reversible modifications, samples were incubated with reducing agent tris(2-carboxyethyl)phosphine (TCEP). We focused on ROS treatment because of greater visibility and reproducibility of the modified precursor bands. In the presence of TCEP, the secondary bands observed in high $H_2O_2$ and DA-treated samples resolved into a single precursor band (Fig. 3c). These results underscore the differences between the two *Msm* DnaB inteins and indicate a reversible responsiveness of *Msm* DnaBi1 to ROS.

**$H_2O_2$ inhibits splicing of DnaBi1 in *M. smegmatis*.** We wanted to ensure that $H_2O_2$ caused growth inhibition of *M. smegmatis*,

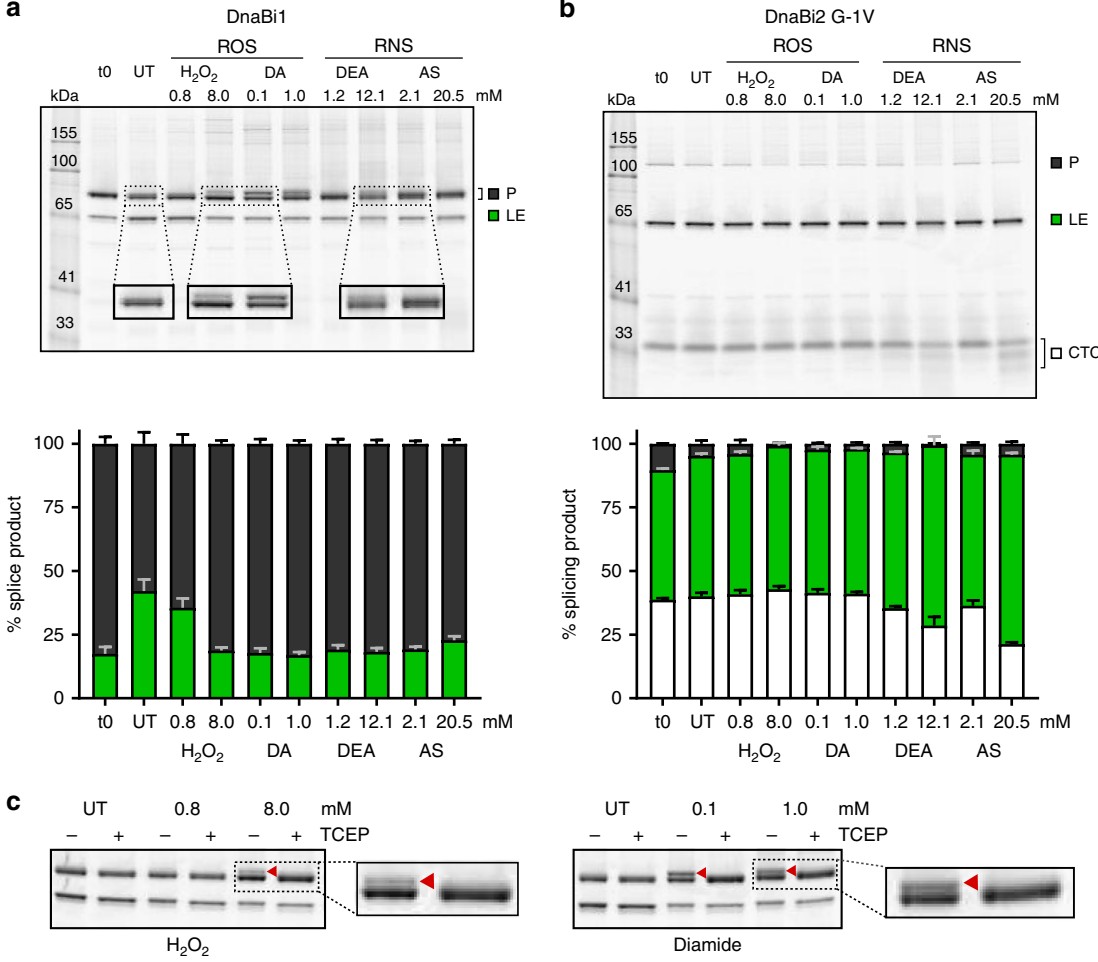

**Fig. 3** *M. smegmatis* DnaB inteins are differentially sensitive to stressors. **a** MIG DnaBi1 accumulates precursor following exposure to stressors. After treatment with ROS and RNS agents there is an increase in the amount of precursor (P) compared to untreated (UT) (top). Additionally, the P band becomes diffuse (RNS) and secondary bands above P are apparent (ROS). The splicing product ratios were quantitated (stack plots below). All the treated samples, except 0.8 mM $H_2O_2$, had increased P compared to UT. DA diamide; DEA DEA NONOate; AS Angeli's Salt. Data are representative of three biological replicates and values are expressed as mean ± s.d. **b** Splicing of MIG DnaBi2 is less responsive to stressors. A mutant version (G-1V) of MIG DnaBi2, which has slower splicing compared to WT (Fig. 2), was evaluated for changes in response to ROS and RNS treatment (top). Unlike MIG DnaBi1, there is no condition that results in precursor accumulation or visible differences in the appearance of the P bands. The splicing product ratios in response to stressors were quantitated (stack plots below). Data are representative of three biological replicates and values are expressed as mean ± s.d. **c** MIG DnaBi1 upper bands are reducible. Reducing agent TCEP was added to UT and ROS-treated samples. The upper bands seen in ROS-treated samples ($H_2O_2$ and DA; red arrowhead) resolved into single precursor bands following treatment, suggesting that a reversible modification is occurring in response to treatment. Data are representative of three biological replicates

which was observed in a concentration-dependent manner (Supplementary Fig. 3). While this does not specifically implicate splicing inhibition, it does confirm that *M. smegmatis* experiences growth arrest following exposure to ROS. Next, we sought to determine if $H_2O_2$ could inhibit DnaBi1 splicing directly in *M. smegmatis*. To accomplish this, we engineered a kanamycin-resistance protein (KanR) fusion with DnaBi1 (Fig. 4a). This is similar to the splicing sensor previously developed using a split intein[29], except we employed KanR Ser154 rather than Ser189 as the +1 nucleophile. This sensor, named "Splice or Die", represents a system to directly measure protein splicing in the host organism where the intein naturally resides.

To ensure that splicing was required for kanamycin resistance, we compared growth of *M. smegmatis* expressing either KanR (no intein), KanR-DnaBi1 wild-type (WT; fusion with active DnaBi1), or KanR-DnaBi1 C118A (fusion with inactive DnaBi1) on media with and without kanamycin (Fig. 4b). We found that while both uninterrupted KanR and the splicing-active KanR-DnaBi1 WT

fusion provide robust resistance to kanamycin, the splicing defective KanR-DnaBi1 C118A does not (Fig. 4b).

Next, we tested the effect of $H_2O_2$ on splicing. To be confident that any reduction in survival was specifically due to splicing inhibition, rather than non-specific killing, we used concentrations of kanamycin and $H_2O_2$ where survival of *M. smegmatis* expressing either KanR or the KanR-DnaBi1 WT fusion were identical (Fig. 4c). We reasoned that under these conditions, any reduction in survival of cells expressing KanR-DnaBi1 compared to KanR must be due to splicing inhibition. Upon treatment with both kanamycin and $H_2O_2$, we observed a selective reduction in survival for *M. smegmatis* expressing KanR-DnaBi1 WT compared to KanR with nine two-fold dilutions, equivalent to 256-fold greater killing after correction for survival differences between the two strains (Fig. 4c). Quantitation of the relative splicing inhibition of KanR-DnaBi1 WT compared to KanR following treatment with both $H_2O_2$ and kanamycin yielded a 213.3 ± 73.9-fold effect (Fig. 4d).

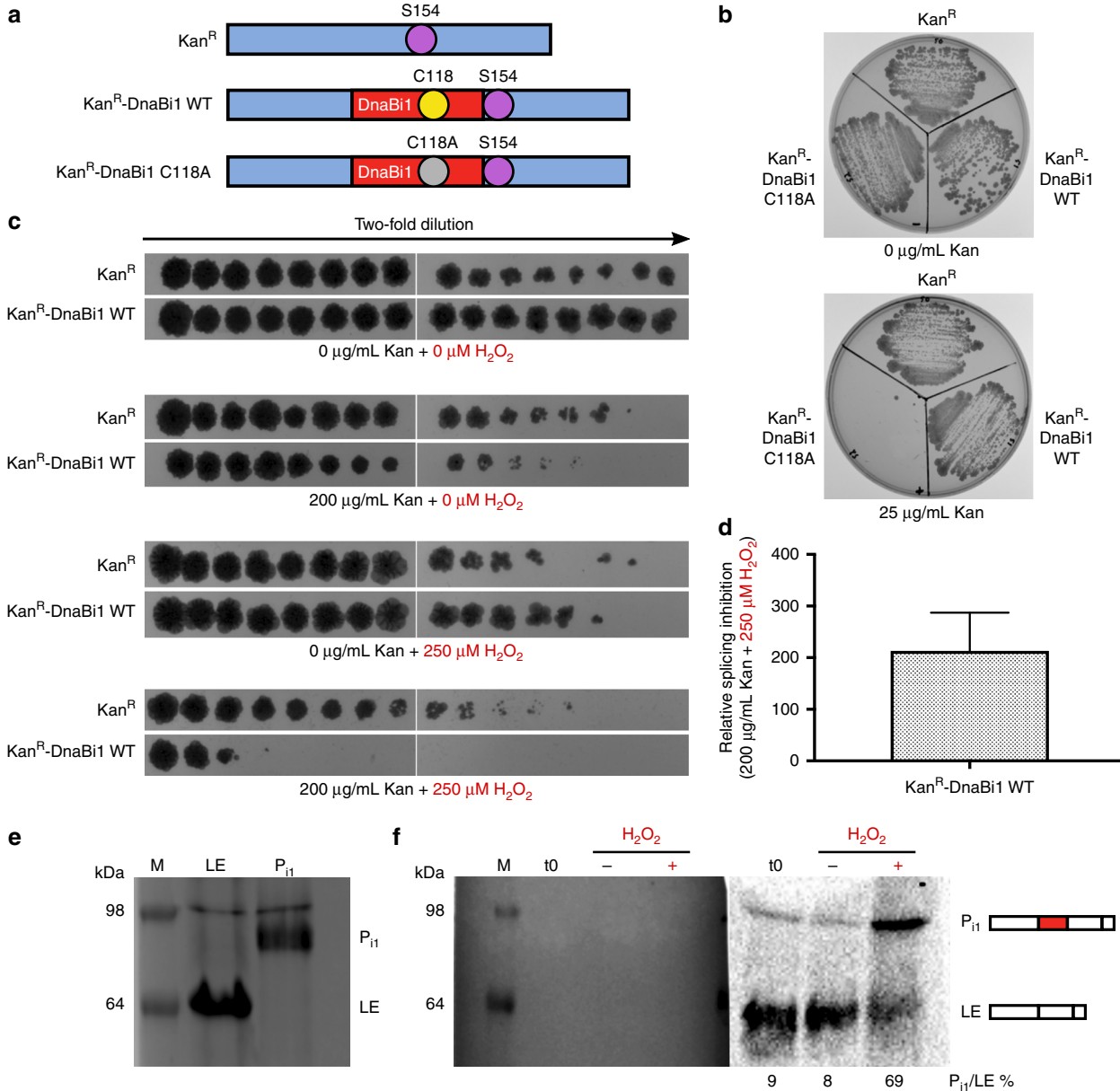

**Fig. 4** DnaBi1 splicing is inhibited by $H_2O_2$ in *M. smegmatis*. **a** Overview of "Splice or Die" constructs. Kanamycin-resistance protein was left uninterrupted ($Kan^R$) or interrupted with either an active DnaBi ($Kan^R$-DnaBi1 WT) or inactive DnaBi with a C118A mutation ($Kan^R$-DnaBi1 C118A). The $Kan^R$ Ser154 serves as the +1 nucleophile for DnaBi1 splicing. **b** DnaBi1 splicing is required to confer resistance to kanamycin in $Kan^R$-DnaBi1 fusion. *M. smegmatis* with $Kan^R$, $Kan^R$-DnaBi1 WT, or $Kan^R$-DnaBi1 C118A (splicing inactive) were spread onto media with either 0 (top) or 25 (bottom) μg/mL kanamycin. Biological replicates ($n = 4$) were performed under similar conditions. **c** Survival of *M. smegmatis* expressing the $Kan^R$-DnaBi fusion is selectively decreased compared to uninterrupted $Kan^R$ in the presence of kanamycin and $H_2O_2$. Two-fold dilutions of early log cells, shown covering a range from ~$3 \times 10^{-1}$ to ~$9.2 \times 10^{-6}$, were spotted onto media with varying concentrations of kanamycin and $H_2O_2$, indicated below. Each experiment consisted of one culture for each strain. Biological replicates ($n = 3$) were performed under similar conditions. **d** Quantitation of the relative splicing inhibition of $Kan^R$-DnaBi1 WT in the presence of kanamycin and $H_2O_2$. The relative splicing inhibition of $Kan^R$-DnaBi1 WT was found to be a 213.3-fold effect. Data are representative of biological replicates ($n = 3$) and are expressed as mean ± s.d. **e** Migration pattern of DnaB ligated exteins and $P_{i1}$ precursor. Overexpressed DnaB ligated exteins, $P_{i1}$ precursor, and a prestained ladder as a size marker (M) were separated by SDS-PAGE and stained by Coomassie. The apparent higher molecular weight of products compared to Fig. 2c is a result of different buffers between the experiments. **f** Western blot analysis shows DnaB $P_{i1}$ precursor accumulates following $H_2O_2$ treatment in *M. smegmatis*. Stationary phase cells (t0) were diluted tenfold in fresh media without ($H_2O_2$, −) or with 5 mM $H_2O_2$ ($H_2O_2$, +) and grown for 1 h. Left, Western blot membrane with prestained ladder post-transfer. Right, chemiluminescence detection of DnaB ligated extein and $P_{i1}$ precursor products, using an anti-DnaB extein 1 antibody. Biological replicates ($n = 3$) were performed under similar conditions

We then asked if DnaB precursor accumulation directly in *M. smegmatis* was detectable by western blot. A probe against extein 1, which detects ligated exteins and precursor products (see Fig. 2c), was used. The gel migration pattern of DnaB ligated extein and $P_{i1}$ precursor products relative to a prestained ladder

was used for band identification (Fig. 4e). In stationary phase cultures, we detected ligated exteins and a small population of $P_{i1}$ precursor (Fig. 4f, t0). Upon outgrowth in the presence of $H_2O_2$ (Fig. 4f, $H_2O_2$, +), we consistently observed accumulation of $P_{i1}$ relative to ligated exteins of four- to eight-fold compared to the

untreated (Fig. 4f, $H_2O_2$, −). This demonstrates intein precursor detection in a native host and precursor accumulation under stress. Together, our results strongly argue for in vivo splicing inhibition of DnaBi1 by $H_2O_2$ and relate in vitro effects directly to those in the native host.

**ROS induces intramolecular disulfide bond in DnaBi1.** To understand the modifications that are occurring on *Msm* DnaBi1 in response to ROS, a mass spectrometry-based (MS) approach was taken. Such modifications can occur via cysteines, of which DnaBi1 has only two. There is Cys118, which serves as the catalytic nucleophile, and Cys48, which is located between splicing blocks A and B (Fig. 1c).

To prevent general oxidation from air, we purified and reduced DnaBi1 aerobically and then ROS-treated reduced intein anaerobically. Addition of $H_2O_2$ or diamide to purified DnaBi1 resulted in the appearance of a band that migrated below the untreated intein (Fig. 5a). This lower band and other treatment-induced differences were reversible by TCEP (Supplementary Fig. 4a). $H_2O_2$ caused increased band diffuseness and the appearance of the lower weight product, which we suspected was an intramolecular disulfide (Fig. 5a). When the $H_2O_2$-treated protein was analyzed by MS, peaks corresponding to an intramolecular disulfide bridge between Cys48 and Cys118 were identified (Fig. 5b), supporting this interpretation. To verify the identity of this peak, tandem MS was performed. Good coverage of the peptide sequences provided additional validation of the fragment's identity (Fig. 5c).

Diamide treatment resulted in similar banding patterns and promoted the appearance of a very high molecular weight product, likely intermolecular disulfide-bonded inteins (Fig. 5a). MS confirmed the presence of a Cys118–Cys118 intermolecular disulfide between Cys118 of two DnaBi1 molecules, as well as showed an intramolecular disulfide and irreversible oxidation events (−$SO_2$, sulfinic acid) on both Cys48 and Cys118 (Supplementary Fig. 4b).

We attempted to generate DnaBi1 mutants unable to disulfide bond but still capable of splicing by mutating Cys48. However, all mutations at this position (C48A/S/T/M/E/D/H) resulted in abrogation of splicing (Supplementary Fig. 5a–e), suggesting that Cys48 is an important residue for activity.

**Conformational flexibility of the catalytic DnaBi1 cysteine.** To understand the mechanism of disulfide bond formation, a crystal structure of the class 3 DnaBi1 intein was solved to 1.95 Å resolution (Fig. 6; Table 1). Class 3 inteins have several unique attributes, most notably three highly conserved residues in splicing blocks B, F, and G, termed the WCT triplet, that characterizes this class (Fig. 6a)[30]. This structure displayed the classic intein shape, with the β-strand fold indicative of the Hedgehog/Intein (HINT) domain (Fig. 6b)[31,32]. *Msm* DnaBi1 lacks a HEN domain, instead having a linker sequence between blocks B and F. The linker, likely a remnant of a lost HEN, was not resolved, typical of linkers at this position in other intein structures[33–35].

The catalytic center of DnaBi1 is composed of several key residues, including the WCT triplet (Fig. 6c, left). The initiating nucleophile Cys118, caught in two distinct orientations, a and b (Fig. 6d), is centrally positioned in the active site and is discussed below. The Gblock Thr137, of the WCT triplet, is positioned to interact with the B block histidine (His65) of the TxxH motif (Fig. 6c, left). The B block His is highly conserved among class 3 inteins[14,30] and has been shown to be crucial for splicing[30,36,37]. DnaBi1 and other class 3 inteins generally lack the conserved threonine of the B block TxxH motif (Fig. 6a). This threonine spring-loads the first position catalytic residue for class 1

inteins[6,38]. Instead, DnaBi1 has an aspartic acid (Asp62), leading to a B block DxxH motif. Its proximity to Ala1 may instead position the N-extein–intein amide bond for attack by Cys118. The WCT B block Trp was predicted to have an architectural role[30] and our structure confirms that WCT Trp67 is part of a core hydrophobic pocket (Fig. 6c, right).

Catalytic Cys118 is centrally positioned in the active site (Fig. 6c, left), allowing interactions with both splice junctions. Structural comparison to the minimized RecA intein[34] indicates that Cys118 is positioned similarly to a highly conserved F block Asp in class 1 and 2 inteins (Fig. 6e and Supplementary Fig. 6), as has been suggested by class 3 intein modeling[30]. This aspartate residue, corresponding to Asp422 in the RecA intein, plays a coordinating role between the reactions at the N- and C-junctions[32,34,39–41].

One of the three DnaBi1 molecules within the asymmetric unit, chain B, showed the thiol sidechain of Cys118 adopting two distinct conformations, a and b (Fig. 6c, d). Gln64 in chain B also displayed a secondary conformation. Conformation a of Cys118 has 36% occupancy and is the unique orientation for this structure, while conformation b has 64% occupancy and is representative of the Cys118 orientation in the two other molecules in the asymmetric unit. Compared to the overall values, the *B*-factors were high for Cys118 with a single conformation in chains A and C of the asymmetric unit, whereas refinement of the dual conformations of Cys118 in chain B significantly reduced the *B*-factor values of this Cys118 (Supplementary Table 1). Together with the finding of two orientations, these results indicate that Cys118 is conformationally flexible. Conformation a is facing up and away from the catalytic center, while conformation b is positioned towards the active site center. However, both Cys118 conformations are distant from Cys48 (Cys118a, 11.0 Å and Cys118b, 12.1 Å; Fig. 6f, inset). MS analysis confirmed a Cys48–Cys118 disulfide bond, therefore other structural shifts are expected to bring the residues within proximity. To understand what these changes might be, we performed modeling to determine how a disulfide bond could form between the two residues. The distance between Cys118a and Cys48 was gradually optimized from 13.0 Å to a reasonable distance for disulfide bond formation, 2.4 Å[42]. The final optimized model showed that the overall intein structure was minimally altered (Fig. 6f). There are small changes around Cys118, but the primary movement occurs on the β-strand containing Cys48. The thiol sidechains for both residues also rotate inward towards each other. This modeling suggests that subtle conformational shifts can bring the cysteines within proximity and implies that the β-strand containing Cys48 is likely responsible for closing that distance.

## Discussion

Inteins are emerging as pervasive post-translational regulatory elements in microbes, where splicing is often coupled to environmental conditions critical to the survival of the host organism or function of the invaded protein[2–5]. Here, we consider the less common scenario of two inteins residing within the same gene, the essential replicative helicase *dnaB* of *M. smegmatis*. While DnaBi1 splices slowly and is inhibited by oxidative and nitrosative insult, DnaBi2 splices rapidly and is unresponsive to the same stressors. Importantly, using our "Splice or Die" reporter to directly measure splicing within a native intein-containing host, we demonstrate that the same oxidative stress inhibits DnaBi1 splicing in vivo. Further, we present in vivo support of DnaBi1 splicing inhibition through detection by western blotting of $P_{i1}$ precursor accumulation in *M. smegmatis* following $H_2O_2$ treatment. Biochemical and structural characterization of DnaBi1

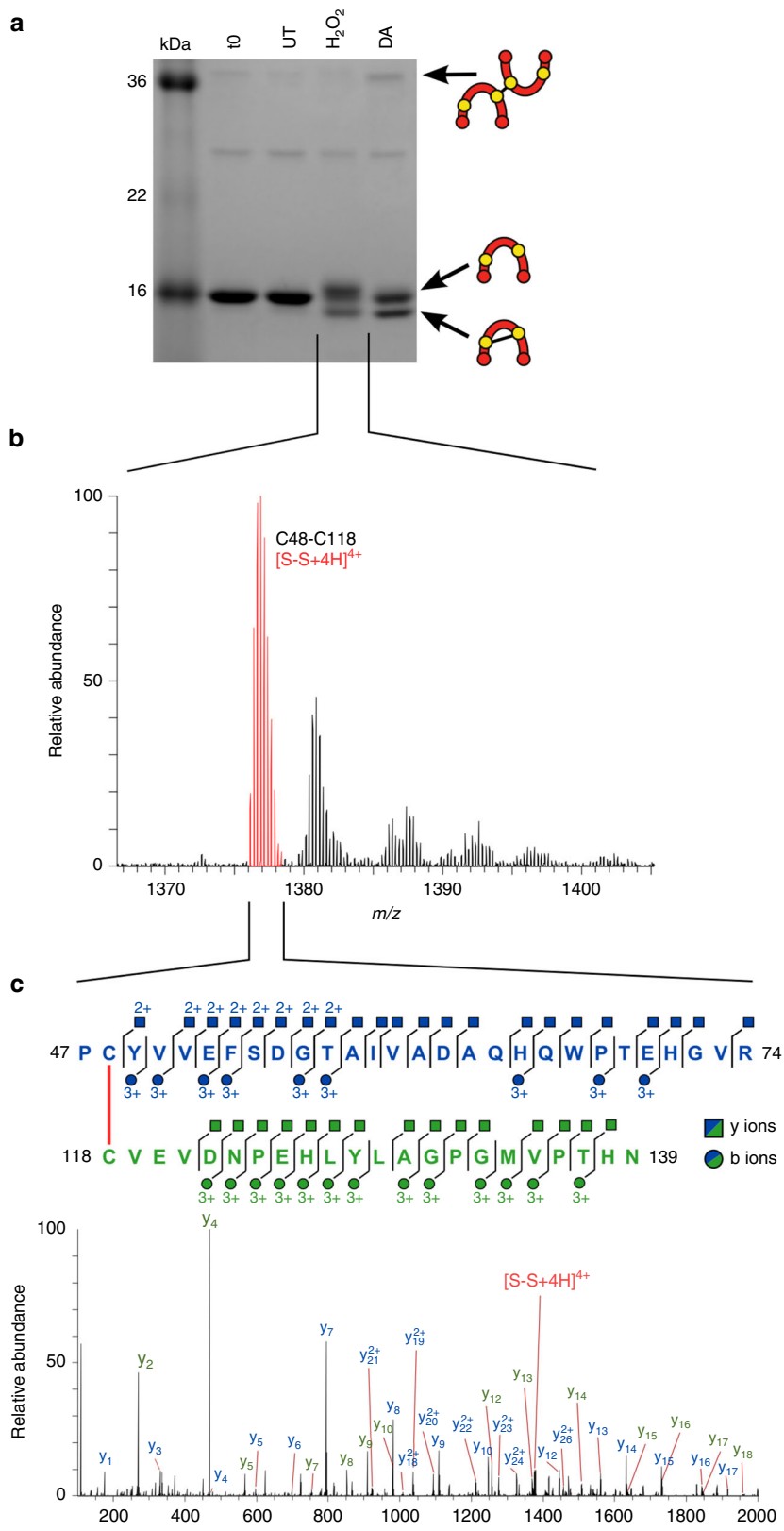

establish that an unusual cysteine is required for splicing and, under oxidative stress, forms an intramolecular disulfide bond with the initiating cysteine nucleophile. We propose a mechanism of splicing-based modulation of DnaB function and, more broadly, the impact on replication fork formation to preserve genome integrity in the presence of physiologically relevant ROS.

The *M. smegmatis* DnaB inteins differ in insertion location, size, and splicing mechanism (Fig. 1), factors that may contribute to the observed differential responses to ROS and RNS. Both

**Fig. 5** DnaBi1 forms a disulfide bond via its catalytic cysteine in response to ROS. **a** DnaBi1 is modified by ROS. Purified DnaBi1 was treated with ROS reagents under anaerobic conditions. Samples were then run anaerobically on a non-reducing SDS-PAGE gel. The identity of the various products is shown schematically as reduced, intra- or intermolecular disulfide-bonded intein. Gel is representative of three technical replicates of experiments prepared for mass spectrometry analysis. UT untreated; DA diamide. **b** Mass spectrometry identifies intramolecular disulfide bond peak. Following $H_2O_2$ treatment, mass spectrometry showed peaks corresponding to an intramolecular disulfide link between fragments containing Cys48 and Cys118, represented here by the most prominent S–S peak at $m/z = 1376.89178^{4+}$ in red. **c** Fragmentation confirmed the peak identity as an intramolecular bond between the two cysteines. The indicated disulfide peak from panel **b** was isolated and tandem mass spectrometry was performed. Fragmentation confirmed the peptide identities, with coverage indicated for both the Cys48-containing peptide (blue) and the Cys118 peptide (green) by the y ions (squares) and b ions (circles). The y series for both peptides are shown on the spectra. The fragmentation ions are colored to match the peptide from which they originated

DnaB inteins utilize cysteine as the initiating nucleophile (Fig. 1c) yet only DnaBi1 is susceptible to cysteine-dependent modification. Not all cysteines are disposed to modifications, as a specific chemical microenvironment is needed to modulate the cysteine $pK_a$ and make it reactive[43]. While catalytic cysteines of inteins generally have low $pK_a$ values to enhance their nucleophilic properties[41,44], the lack of response with class 1 DnaBi2 indicates that splicing inhibition through cysteine-based redox is not a universal phenomenon. This is the case even within class 1 inteins, where the three inteins present in SufB, RecA, and DnaB of *M. tuberculosis* displayed distinct splicing behaviors and different sensitivities to oxidation in vitro[3].

Differences between the two DnaB inteins could allow separate utilization of the two inteins by the host, enabling *M. smegmatis* to respond to a diverse set of stress conditions. Interestingly, the fast-splicing, unresponsive intein, DnaBi2, contains a HEN, whereas the responsive intein, DnaBi1, does not. This observation is consistent with the idea that HEN-containing inteins are usually active, mobile, and engaged in the parasitic intein homing cycle[8,45], whereas inteins that have lost the HEN domain and mobile properties must rely on alternative approaches to maintenance. The removal of the intein sequence without inactivating the host protein is difficult but may suffice for long-term intein survival. Alternatively, the intein may become adapted to the host and thus be maintained by serving a function[2,4,46].

While inteins are abundant in archaea and bacteria[10], they are often found in non-model systems, making splicing studies in the native organism challenging. As such, we have lacked in vivo evidence of protein splicing modulation in the native host organisms to correspond to the rapidly growing examples of conditional protein splicing in vitro. Our "Splice or Die" reporter has allowed direct and quantitative monitoring of the splicing process in the native host. Further in vivo support is provided by detection of DnaB $P_{i1}$ accumulation following $H_2O_2$ treatment by western blot. We thus demonstrate that DnaBi1 splicing can be inhibited in the native *M. smegmatis* host in response to the same oxidative stress (Fig. 4) shown to block splicing in vitro. We believe these results represent a landmark step in our efforts to understand splicing regulation in the natural intein context and provide an important proof of principle that in vitro measures of protein splicing can likely translate to natural systems.

The class 3 intein structure enhances our understanding of an atypical splicing mechanism that has been biochemically characterized[30]. The central position of Cys118 between the splice junctions harkens to that of a conserved Asp found in class 1 and 2 inteins (Fig. 6 and Supplementary Fig. 6), a residue shown to coordinate the N- and C-terminal splicing reactions. This results in a more compact catalytic center and suggests that Cys118 may participate in the splicing chemistry at both termini. This could be facilitated by the crystallographically determined conformational flexibility of Cys118 (Fig. 6; Supplementary Table 1) considered below.

DnaBi1 lacks the spring-loading threonine of the B block TxxH motif present in class 1 inteins[38], having an aspartate instead (Asp62) (Fig. 6c, e). Asp62 is not positioned to prime Cys118 and the centralized location of Cys118 may not require the same mechanism needed for class 1 inteins. Instead, Asp62 may serve a stabilizing or activating role during the first steps of splicing, as has been previously observed with the canonical Thr in a class 2 intein, also lacking a position 1 nucleophile[40]. A key role for a B block Asp in class 3 inteins is also supported by mutagenesis, resulting in increased C-terminal cleavage[30].

Beyond the biological insights provided by our class 3 intein structure, we are excited by the technologies that may develop as a result. Class 1 inteins have been utilized extensively in protein engineering[35,47,48] and we expect that class 3 inteins hold untapped potential in this arena, particularly since the catalytic residue arrangement is compacted compared to class 1. Further, DnaBi1 can form a natural redox trap, housed entirely within the intein, which may be useful to control splicing in non-native systems. The structural insight gained from this class 3 intein may provide a scaffold in which to engineer and design new splicing-based technologies.

The oxidation of DnaBi1 resulted in an intramolecular disulfide bond, validated by mass spectrometry, between catalytic Cys118 and non-conserved Cys48 (Fig. 5). Examples of redox-based regulation in inteins have shown cysteines in conserved splicing blocks[23] and exteins[3,6] as disulfide bonding partners. Cys48 is not in a splicing block or extein sequence yet is pivotal to both catalysis and splicing regulation. It is unclear how Cys48 influences splicing, although it does not appear have a direct role in catalysis (Supplementary Fig. 5). Residues outside of the conserved splicing blocks influence splicing of other inteins[34,47–49], but it is intriguing that a non-conserved residue is important for both splicing and splicing modulation.

The conformational freedom of Cys118, shown crystallographically (Fig. 6; Supplementary Table 1), appears to promote disulfide bonding with Cys48, in addition to disposing the residue towards both splice sites, and suggests that Cys118 acts as a toggle between a splicing active and inactive state in a redox-dependent fashion (Fig. 7a). The use of a thiol-based redox sensor is not uncommon in prokaryotes[50] and has begun to emerge as a theme among inteins from a diverse set of microbes[3,6,23]. Such cysteine-based switches are important for responding to oxidative stressors and protecting proteins during adverse conditions[51].

As the DnaB protein is the replicative helicase within mycobacteria, its function is essential for replication and growth. Under oxidative stress, arrest of replication may be advantageous to preserve DNA integrity much as it appears to be in mammalian systems[52]. We propose a model where the full precursor ($P_{i1,i2}$) is translated and DnaBi2 rapidly excises itself, leaving an alternative precursor ($P_{i1}$) with DnaBi1 still present (Fig. 7a). Under oxidizing conditions, an intramolecular disulfide forms between Cys48 and Cys118, inhibiting splicing and trapping $P_{i1}$ in an inactive state. Return to a reducing environment resolves

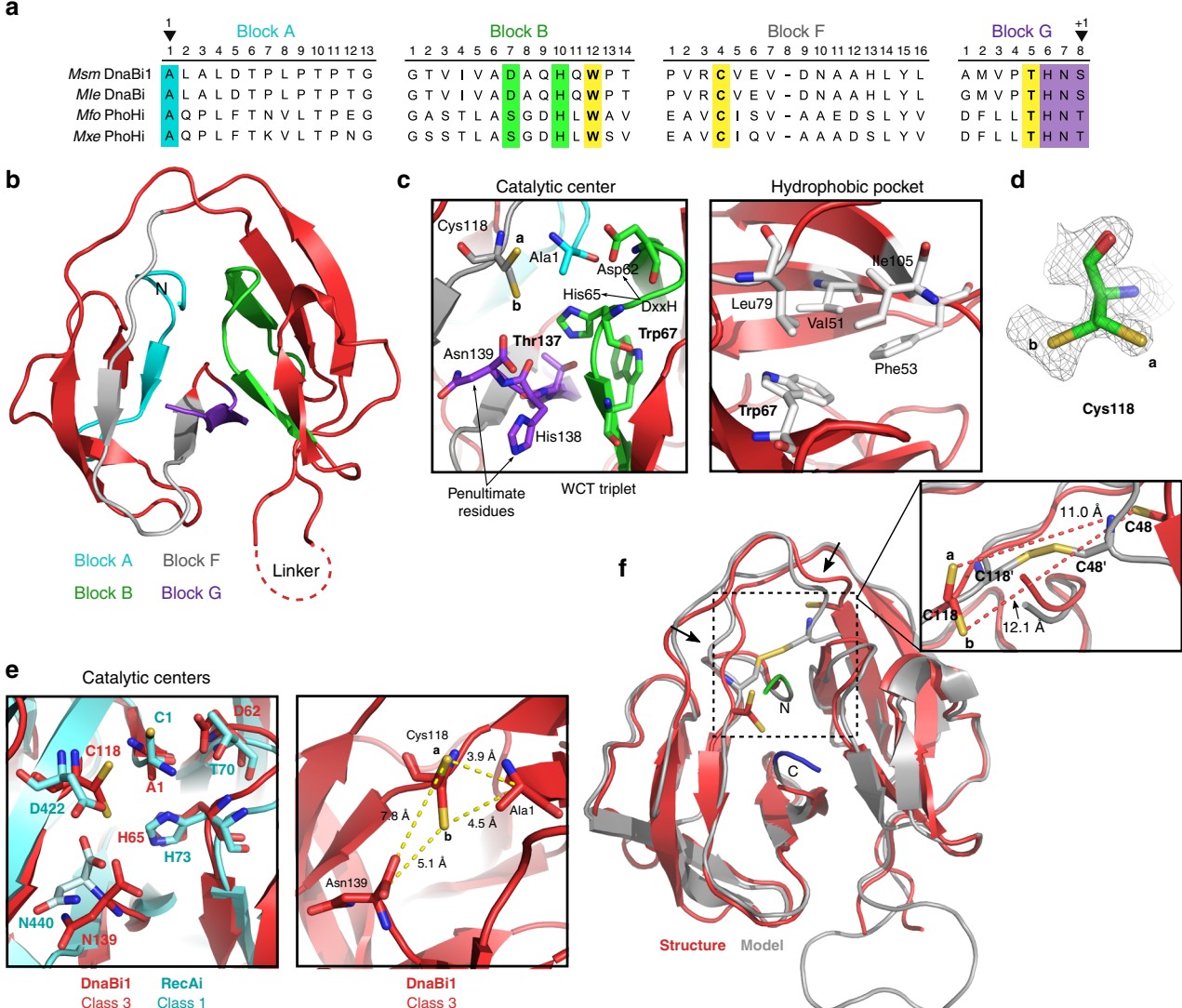

**Fig. 6** Structure reveals conformational variability of Cys118. **a** Mycobacterial class 3 intein alignment. Four class 3 mycobacterial inteins from two intein-containing proteins, DnaB and PhoH (phosphate starvation-inducible protein) show the four splicing blocks and class 3 features. The WCT triplet residues are yellow. Important residues are highlighted, and the 1 and +1 residues noted (arrowheads). *Msm M. smegmatis; Mle M. leprae; Mfo M. fortuitum; Mxe M. xenopi.* **b** DnaBi1 structure provides insight into class 3 inteins. The class 3 intein crystal structure was solved to 1.95 Å. The four splicing blocks are in cyan (block A), green (block B), gray (block F), and purple (block G). Amino (N) terminus is annotated. **c** Class 3 intein structural features. The catalytic center is shown (left). The WCT triplet residues (Trp67, Cys118, Thr137) are indicated. Cys118 shows two orientations, a and b, where a faces away from the catalytic center and b is oriented towards it. Important residues include the B block DxxH motif (Asp62 and His65) and the G-block penultimate His138 and terminal Asn139. The hydrophobic pocket containing Trp67 is presented (right) and hydrophobic packing residues are indicated (white). **d** Electron density map of Cys118 showing the distinct a and b orientations. **e** Overlay of class 3 and class 1 active sites. Residues involved in the splicing mechanism for both intein classes are shown (left). Class 3 DnaBi1 residues are red and class 1 RecAi residues are cyan. Cys118 is in the same location as Asp422, a residue proposed to coordinate the N- and C-junction reactions[34, 41]. The right panel shows distances between centrally located Cys118 to the N- and C-extein junctions. **f** DnaBi1 structure and an optimized disulfide-bonded model overlay show minor conformational differences. The structure (red) was overlaid with a model (gray) optimized for a disulfide linkage between Cys48 and Cys118. The a and b catalytic Cys118 conformations are too distant from Cys48 for an intramolecular disulfide bond (a 11.0 Å; b 12.1 Å; inset). The disulfide-bonded model undergoes minor structural changes (arrows) and includes two N- (green) and C-extein (blue) residues

the disulfide bond and results in instantaneous initiation of protein splicing, allowing DnaB protein to assemble at the replication fork and conduct ATP hydrolysis (Fig. 7b).

It has been shown for another intein, inserted in the P-loop of *Pyrococcus horikoshii* RadA, that the intein's presence disrupts ATPase function[4], and it is reasonable to assume this would be true for P-loop-inserted DnaBi1. However, it is unclear if full precursor or partially spliced products, like $P_{i1}$, may retain partial DnaB activity, such as DNA binding. Recent work has shown that

the intein-containing RadA precursor is able to bind ssDNA, its native substrate[2]. Similarly, it is possible that certain DnaB functions, such as dimerization or DNA binding, could occur with the DnaBi1-containing alternate precursor. In a broader context, if the helicase cannot assemble, replication would be stalled. In human cells, ROS has been shown to slow replication fork progression and thereby protects the genome from DNA damage[52]. Likewise, it is tempting to speculate that in *M. smegmatis* ROS could inhibit intein splicing, causing delays in

## Table 1 Data collection and refinement statistics

|  | Native[a] | Se-Met[a] |
|---|---|---|
| **Data collection** |  |  |
| Space group | P2$_1$ | P2$_1$ |
| Cell dimensions |  |  |
| $a$, $b$, $c$ (Å) | 64.05, 56.91,64.73 | 63.65, 56.83, 64.55 |
| $\alpha$, $\beta$, $\gamma$ (°) | 90.00, 106.60, 90.00 | 90.00, 105.71, 90.00 |
| Resolution (Å) | 38.2–1.95 (2.02–1.95)[b] | 41.9–2.49 (2.58–2.49) |
| $R_{sym}$ or $R_{merge}$ | 0.095 (0.754) | 0.11 (0.34) |
| $I$ /σ$I$ | 11.9 (2.1) | 13.9 (4.5) |
| Completeness (%) | 97.0 (97.9) | 99.0 (93.1) |
| Redundancy | 2.7 (2.3) | 3.7 (3.8) |
| $R_{ano}$ |  | 0.12 |
| Se sites |  | 12 |
| Figure of merit |  | 0.39 |
| **Refinement** |  |  |
| Resolution (Å) | 38.2–1.95 (2.02–1.95) |  |
| No. reflections | 31,744 (3171) |  |
| $R_{work}$/$R_{free}$ | 0.203/0.236 |  |
| No. atoms |  |  |
| Protein | 2861 |  |
| Ligand/ion | 0 |  |
| Water | 303 |  |
| $B$-factors (Å$^2$) |  |  |
| Protein | 31.5 |  |
| Ligand/ion |  |  |
| Water | 39.4 |  |
| R.m.s. deviations |  |  |
| Bond lengths (Å) | 0.008 |  |
| Bond angles (°) | 1.000 |  |

[a]One crystal was used for each structure [b]Values in parentheses are for highest-resolution shell

replication fork assembly, thereby protecting the cell from replication stress and possibly leading to a dormant state[53]. Splicing would restore the integrity of the fork and replication could proceed immediately when favorable reducing conditions return (Fig. 7b). This type of splicing inhibition would provide an immediate, post-translational response to adverse environmental conditions. Instantaneous restoration of function and replication restart could be achieved by dissolution of the disulfide bond under reducing conditions, when the ROS-dependent stress is relieved.

## Methods

**DnaB extein modeling and class 3 intein sequence alignment**. The *M. smegmatis* (*Msm*) DnaB ligated exteins model (Fig. 1c) was generated by submission of the protein sequence to I-TASSER[54]. The DnaB ligated exteins model had a C-score of −0.22 and an estimated RMSD of 7.6 ± 4.3 Å. Sequences for Fig. 6a were accessed through NCBI and accession numbers listed: *M. smegmatis* DnaBi1 (GenBank: AFP43135); *M. leprae* DnaBi (GenBank: CAA17948); *M. fortuitum* PhoHi (GenBank: OBK00716); *M. xenopi* PhoHi (GenBank: EID13626).

**Bacterial strains and growth conditions**. All strains used in the present study can be found in Supplementary Table 2. *Escherichia coli* DH5α (Gibco-BRL), MG1655 (DE3) (James Imlay), BL21(DE3) (Novagen), JM109 (Stratagene), B834(DE3) (Novagen), and MC1061[28] were grown in Luria Broth (LB), unless otherwise indicated, with aeration at 250 rpm. Media contained kanamycin (50 μg/mL), chloramphenicol (25 μg/mL), or carbenicillin (50 μg/mL) where appropriate. Plasmids were transformed into cells by electroporation using a Bio-Rad Gene Pulser and recovered for 1 h at 37 °C in SOC medium (0.5% yeast extract, 2% tryptone, 10 mM NaCl, 2.5 mM KCl, 10 mM MgCl$_2$, 10 mM MgSO$_4$, and 20 mM glucose). Transformants were selected by plating on LB agar with the appropriate antibiotic and incubated at 37 °C overnight.

**Construction of plasmids**. All plasmids used in the present study can be found in Supplementary Table 2 and all oligonucleotides, synthesized by Integrated DNA

Technologies (IDT), are in Supplementary Table 3. Plasmid DNA was prepared using E.Z.N.A. Plasmid Mini Kit (Omega). DNA was visualized in 1% agarose gels using EZ-Vision DNA Dye (Amresco). PCR fragments were amplified using CloneAmp HiFi PCR Premix (Clontech) from *Msm* mc$^2$ 155, *Mtu* H37Rv, or *Mle* Br 4923 genomic DNA (BEI Resources). Digest and PCR fragments were gel purified using Zymoclean Gel DNA Recovery Kit (Zymo Research). Restriction enzymes (NEB), T4 ligase (NEB), and In-Fusion HD Cloning Plus Kit (Clontech) were all used per manufacturer protocol. Mutagenesis was performed using QuikChange Lightning Site-Directed Mutagenesis Kit (Agilent) for single amino acid mutations or QuikChange Lightning Multi Site-Directed Mutagenesis Kit (Agilent) for multiple amino acid mutations. All clones were verified by sequencing (EtonBio).

**In vitro MIG splicing assay**. To monitor splicing of the DnaB inteins, maltose-binding protein (MBP)-intein-green fluorescent protein (GFP) (MIG) reporter constructs were made (Supplementary Table 2). MIG plasmids were transformed into MG1655(DE3) as described in "Bacterial strains and growth conditions". Cells were subcultured 1:100 from overnight cultures into fresh LB, grown at 37 °C to an OD$_{600}$ of 0.5, and induced with 0.5 mM IPTG for 1 h at 30 °C. Samples were pelleted for 10 min at 4000 rpm at 4 °C and lysed using a tip sonicator in lysis buffer (50 mM Tris pH 8.0, 10% glycerol). A t0 sample was taken and the sample lysate was incubated at 30 °C for the duration of the experiment. For splicing time courses, aliquots were removed at the indicated times. For ROS/RNS treatments, the indicated compound was added to lysate at the desired concentration immediately prior to incubation. MIG DnaBi1 samples were incubated for 5 h and MIG DnaBi2 G-1V samples were incubated for 2 h. To assess reversibility of potential modifications and secondary bands, samples were treated with 40 mM TCEP for 10 min on ice. Samples were run under non-reducing conditions on Novex WedgeWell 12% Tris-Glycine gels (Invitrogen) using loading dye lacking β-mercaptoethanol and a Typhoon 9400 scanner (GE Healthcare) was used to visualize GFP-containing products. Quantitation and analysis were done using ImageJ and GraphPad Prism (v7.02). Uncropped images of gels are provided in Supplementary Figure 7.

**Splicing in native DnaB exteins**. The full *dnaB* gene from *Msm* mc$^2$ 155 was amplified from genomic DNA and cloned by InFusion (Clontech) into pET47b (Novagen). Splicing-inactive versions were made by site-directed mutagenesis (Agilent) and inteinless versions were made by splicing by overlapping extension (SOEing) PCR (see Supplementary Table 3). BL21(DE3) cells containing wild type and mutant pET47b DnaB constructs were subcultured 1:100 from overnight cultures into fresh LB, grown to OD$_{600}$ 0.6, and induced overnight at 16 °C with 0.5 mM IPTG. Pellets were lysed in lysis buffer (20 mM Tris pH 8.0, 500 mM NaCl, 10% glycerol) and whole-cell lysate was analyzed by sodium dodecyl sulfate polyacrylamide gel electrophoresis (SDS-PAGE) on 8–16% Mini-PROTEAN TGX Precast Protein Gels (Bio-Rad) for splicing. Uncropped images of gels are provided in Supplementary Figure 7.

**FRET assay**. MC1061 cells containing CFP-intein-YFP (CIY) DnaBi1 constructs were grown to OD$_{600}$ ~0.5 and induced with 0.2% arabinose for 6 h at 25 °C. Cells were pelleted and prepared for FRET analysis by lysing with B-PER (Thermo-Fisher). Samples were pelleted, and soluble extract was transferred to a 96-well microtiter plates for measurements. Samples were excited at 400 nm and emission readings for CFP (485 nm) and FRET/YFP (540 nm) were taken every 5 min for a 1055 min (~17.5 h) time course at 37 °C using a BioTek Synergy H1 plate reader. Either 100 mM hydroxylamine (HA) or 20 mM dithiothreitol (DTT) was added as an external nucleophile to induce N-terminal cleavage. Samples were run in duplicate and reads averaged. The FRET ratio of each sample was determined, normalized, and plotted using GraphPad Prism (v7.02). Uncropped images of gels are provided in Supplementary Figure 7.

**M. smegmatis Kan$^R$-DnaBi1 fusion constructs**. *Msm dnaB* intein 1 was inserted in-frame using Gibson Assembly (NEB) into *kan$^R$* (aminoglycoside O-phospho-transferase APH(3′)-Ia; NCBI Reference Sequence: WP_000018329.1). Kan$^R$-DnaBi1 fusions were screened to identify candidates that required splicing for survival on kanamycin-containing media. Kan$^R$-DnaBi1 fusion with Ser154 of Kan$^R$ as the +1 nucleophile was chosen and subsequently subcloned into a mycobacterial shuttle vector, pMBC283, using Gibson Assembly (NEB). *M. smegmatis* was grown in standard 7H9 (liquid) and 7H10 (solid) media. However, for spot titer studies involving H$_2$O$_2$, 7H10 was prepared without catalase and albumin. Concentrations of cells, H$_2$O$_2$, and kanamycin used are described in the figure. For spot titers, 1.25 μL of cells at each titer were spotted and grown at 37 °C for 5 days.

**Western blotting**. Wild-type *Msm* mc$^2$ 155 was grown to stationary phase for ~4 days at 37 °C in Middlebrook 7H9 media, lacking catalase and albumin, with 50 mg/L carbenicillin (t = 0 sample). Cells were then diluted tenfold in fresh Middlebrook 7H9 media, again lacking catalase and albumin, in the presence or absence of 5 mM H$_2$O$_2$ for 1 h. Samples were lysed using Yeast Protein Extraction Reagent (Thermo Scientific), separated on 12% SDS-PAGE gels, transferred to a

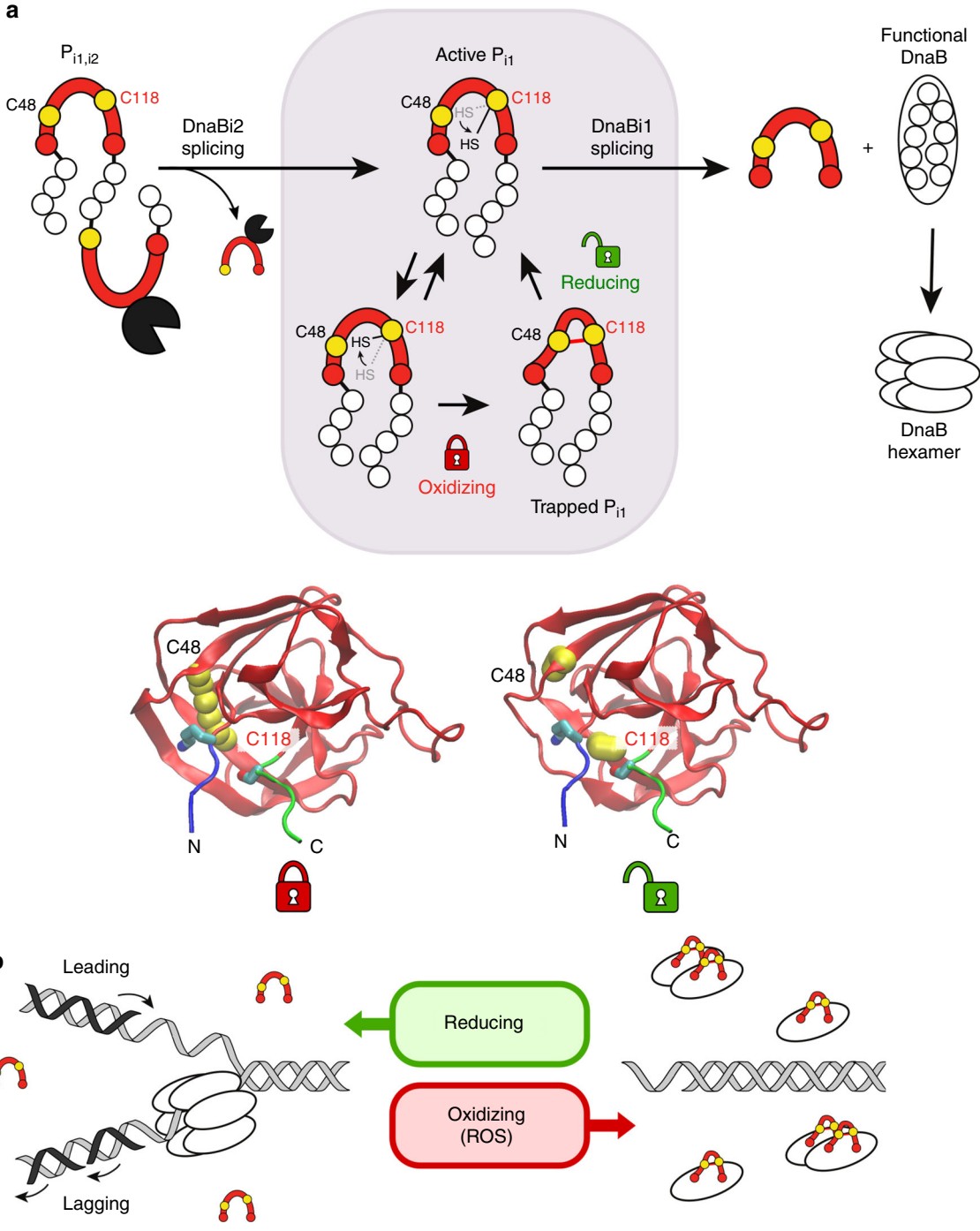

**Fig. 7** Model for impact of ROS/RNS stress on DnaB splicing and DNA replication. **a** Model for DnaBi1 splicing regulation by ROS. The full-length precursor (P$_{i1,i2}$) is expressed and DnaBi2 rapidly splices out, leaving an active alternative precursor (P$_{i1}$) with DnaBi1 still present (shaded box). Cys118 has conformational freedom, alternating between the a and b orientations (arrow). In an oxidizing environment, such as that resulting from ROS, an intramolecular disulfide bond forms between Cys48 and catalytic Cys118 of DnaBi1 (red line), locking P$_{i1}$ in a splicing-inactive state. Once the cell restores a reducing environment, the disulfide bridge is resolved and Cys118 toggles to initiate splicing, whereupon DnaB is immediately functional and hexamerizes to assume its role in replication. The structural changes are shown below in two DnaBi1 models with several extein residues (N-extein, blue; C-extein, green), indicating the movement of Cys118 from a disulfide-bonded state with Cys48 to a reduced state, where Cys118 is positioned to initiate splicing. **b** Model for replication arrest through splicing modulation. In the presence of ROS, intein splicing is inhibited through cysteine oxidation (red horseshoe as in panel **a**). This prevents some, but not necessarily all, DnaB functions and prevents replication fork formation (right). When the environment becomes favorable, the cysteines are reduced, enabling DnaBi1 splicing to proceed. This produces fully active DnaB protein, which is able to participate in replication fork formation and progression

PVDF membrane (BioRad Trans-Blot Turbo transfer system), and probed for DnaB extein 1 at a dilution of 1:3500 (Covance; anti-rabbit antibody NY1872). HRP-conjugated goat anti-rabbit secondary antibody (Advansta) at a dilution of 1:10,000 was used, and signal was detected by chemiluminescence (Li-COR). The identity of DnaB ligated extein and precursor $P_{i1}$ bands was verified by comparing their migration patterns relative to a prestained ladder, as well as ligated extein and precursor $P_{i1}$ overexpressed in *E. coli*. Uncropped images of the western blot are provided in Supplementary Figure 7.

**DnaBi1 purification and treatment**. DnaBi1 plus four native N-extein residues was cloned into pXI[55], creating a chitin-binding domain (CBD)-DnaBi1 fusion protein. JM109 cells containing pXI DnaBi1 were subcultured 1:100 from an overnight culture into fresh LB and grown to $OD_{600}$ 0.6. Cells were induced overnight with 0.5 mM IPTG at 16 °C. Cells were harvested and lysed by tip sonication in chitin-binding buffer (CBB: 20 mM Tris pH 8.5, 200 mM NaCl, 0.1 mM EDTA). Lysate was centrifuged to separate the soluble fraction and added to washed chitin beads (NEB). Sample was incubated overnight at 4 °C with nutation. Beads were then washed in CBB to remove unbound protein. To elute the DnaBi1 intein from beads, 200 mM DTT in CBB was added and incubated at 4 °C for 2 days with nutation. An additional 100 mM DTT booster was added to the sample and allowed to incubate overnight at 4 °C. Eluate was collected by gravity column and incubated with fresh chitin beads overnight at 4 °C to bind any remaining CBD-containing precursor. The final eluate was collected by gravity column and concentrated in a 3K MWCO Amicon filter (Millipore).

Purified DnaBi1 was treated with 40 mM TCEP and brought into an anaerobic chamber (Coy). The protein was exchanged into anaerobic exchange buffer (20 mM Tris pH 7.5, 200 mM NaCl) using 7K MWCO Zeba spin desalting columns (Thermo) to remove TCEP. The protein was diluted to a final concentration of 10 μM and treated with 1 mM $H_2O_2$ or diamide at 30 °C for 15 min. A portion of sample was removed for gel analysis. Samples were combined with non-reducing loading dye lacking β-mercaptoethanol and run anaerobically on a Novex WedgeWell 16% Tris-Glycine gels (Invitrogen). The remaining sample was processed for mass spectrometry analysis. Uncropped images of gels are provided in Supplementary Figure 7.

**Mass spectrometry and data analysis**. Treated DnaBi1 protein was denatured with 6 M urea at 37 °C for 30 min in the anaerobic chamber. The urea concentration was diluted to ~0.8 M with 50 mM Tris pH 7.6, 1 mM $CaCl_2$. Activated trypsin (Promega) was then added to a final ratio of 1:20 (trypsin:DnaBi1) and incubated overnight at 37 °C. Samples were removed from the anaerobic chamber, desalted using Pierce C18 spin columns (Thermo), and eluted in 70% acetonitrile (LC-MS grade; Pierce) in water (LC-MS grade; Fluka Analytical). Acetonitrile was removed by speed-vacuum centrifugation and samples were lyophilized. Mass spectrometry solvent (150 mM ammonium acetate pH 7.0) was prepared with LC-MS grade $NH_4OAc$ (Sigma-Aldrich) and LC-MS grade water (Fluka Analytical). Lyophilized samples were reconstituted in 150 mM ammonium acetate and were further diluted 1:10 in 150 mM ammonium acetate and 10 % isopropanol just prior to analysis.

Digests were analyzed by direct infusion electrospray ionization (ESI) on a ThermoFisher Scientific LTQ Orbitrap Velos mass spectrometer running in positive ion mode. All analyses were performed in nanoflow ESI mode with quartz emitters produced on a Sutter Instruments Co. P2000 laser pipette puller. Sample was loaded into the back of an emitter and a stainless-steel wire was inserted to supply an ionization voltage of 1.0 kV. High-resolution analysis was performed by calibrating the Orbitrap mass analyzer with a solution of 1 mg/mL cesium iodide in 50% methanol over a range of 500–3000 $m/z$ with up to 1 ppm mass accuracy. Tandem mass spectrometry (MS/MS) was accomplished by isolating precursor ions of interest in the curved linear ion trap (C-trap) element, activating fragmentation in the higher-energy collisional dissociation cell, and detection in the Orbitrap. High-resolution full scan and fragmentation data peak lists were processed by Xcalibur 2.1 software (Thermo). In silico tryptic digest and MS/MS peptide sequencing were processed with the Protein Prospector v5.20.0 programs MS-Bridge and MS-Product (http://prospector.ucsf.edu).

**Selenomethionine labeling and purification of DnaBi1**. B834(DE3) cells containing pXI DnaBi1 were grown in SelenoMethionine Medium Complete (Molecular Dimensions) supplemented with 1× methionine overnight at 37 °C. Cells were washed three times in SelenoMethionine Medium Complete lacking methionine or selenomethionine, resuspended, and subcultured 1:50 into fresh media containing 1× selenomethionine. Cells were grown to an $OD_{600}$ ~0.6 and induced as described above for pXI. Following induction, cells were harvested and lysed. Protein was purified by batch chitin purification as described above. Purified protein was then passed over a Superose 12 10/300 GL column (GE Healthcare Life Sciences) on an AKTA Pure (GE Healthcare Life Sciences) to separate out additional impurities. Eluate was collected and concentrated to 12 mg/mL in a buffer containing 20 mM Tris, pH 8.0, 200 mM NaCl, 1 mM TCEP for crystallization.

**Crystal structure determination**. The affinity-purified native DnaBi1 intein was further purified by a gel filtration 16/60 Superdex 75 column (GE HealthCare).

The fractions containing intein were pooled and concentrated to 13 mg/mL in a buffer composed of 20 mM Tris, pH 8.0, 200 mM NaCl, 1 mM DTT. Initial crystallization conditions were established by screening the Hampton Research crystallization screen I, II, and index HT, using the hanging-drop vapor diffusion method. Upon optimization, large crystals were grown at room temperature in hanging drops, by mixing 1 μL of DnaBi1 and 1 μL of reservoir solution containing 20% (v/v) 2-methyl-2,4-pentanediol (MPD), 0.1 M sodium acetate, pH 4.6, 0.2 M sodium chloride (NaCl). The crystals of DnaBi1 belong to space group $P2_1$, with unit cell parameters of $a = 64.05$ Å, $b = 56.91$ Å, $c = 64.73$ Å, $\beta = 106.6°$ and three molecules per asymmetric unit cell. The Se-Met DnaBi1 was crystallized in a similar condition with a modified reservoir solution composed of 0.1 M sodium acetate, pH 4.6, 32% MPD, 0.2 M NaCl, 3% glycerol, 1% PEG4000.

Prior to data collection, all crystals were transferred to a cryo-protectant solution containing crystallization buffer with an MPD concentration of 30%. The crystals were flash-cooled directly in liquid nitrogen. Diffraction data for the DnaBi1 crystals were collected at 100 K at the beamline 14-1 of the Stanford Synchrotron Radiation Lightsource (SSRL). Native and Se-Met data were collected at 1.18076 and 0.97919 Å, respectively. Data were processed, scaled, and reduced using the programs HKL2000 (ref. [56]) and Phenix suite[57].

The structure of the DnaBi1 intein was determined using single anomalous dispersion phasing method with the Phenix program suite[57]. The model of a DnaBi1 monomer was completed by manually fitting the electron density map with the DnaBi1 sequence using the program TURBO FRODO[58] and Coot[59]. The other two molecules were generated through non-crystallographic symmetry. The structure refinement was carried out using the Phenix program suite[57] with a final $R_{work}$ of 20.3% and a $R_{free}$ of 23.6% (Table 1). In the final structure, 97.6%, 2.4%, and 0% of residues fall into favored, allowed, and disallowed regions in Ramachandran plot.

**Modeling of intramolecular disulfide-bonded structures**. Disulfide bond formation between Cys48 and Cys118 was modeled in all four intein conformations captured in the crystal structure, two of which are identical except for an alternate orientation of Cys118. The CHARMM program[60], version c35b3, was used for modeling with the CHARMM26 additive force field for proteins[61]. All non-hydrogen atoms in protein residues other than nearest neighbor residues for the cysteines (i.e. residues 47–49 or 117–119) were constrained by a harmonic constraint with a force constant of 1 kcal/mol/Å². All non-bonded interactions were included by using a cutoff of 999.0 Å, and all bonds to hydrogen atoms were constrained using the SHAKE algorithm[62]. A distance restraint was imposed between the sidechain sulfur atoms (SG) of Cys48 and Cys118 with a force constant of 1000 kcal/mol/Å². The minimum of this restraint was changed from 13 to 3 Å in 0.5 Å decrements, with an optimization at each minimum consisting of 1000 steps of Langevin dynamics at 300 K with a friction coefficient on all non-hydrogen atoms of 60 ps$^{-1}$, followed by Steepest Descent (SD) minimization[63] of 5000 steps with an energy tolerance of 0.001 kcal/mol. A patch to form the disulfide bond was applied to the final minimized structure, and the structure was optimized further using 5000 SD minimization steps, 1000 Langevin dynamics steps, and another 5000 SD minimization steps to obtain the predicted intramolecular disulfide-linked structure.

**DnaBi1 model validity assessment**. The disulfide-linked model was assessed using PROSA-Web[64], MolProbity[65], and Verify3D[66]. Its PROSA-Web Z-score is −4.86, its MolProbity score is 2.43 (roughly compares to X-ray resolution), and Verify3D indicates that 93% of its residues have averaged 3D-1D scores ≥0.2 (80% are required for a "pass" score). Additional details from the MolProbity analysis are shown in Supplementary Table 4.

**Statistical analysis**. Values with error bars represents the mean ± standard deviation.

## Data availability

All data are presented in the manuscript and supporting materials are available upon reasonable request from the corresponding authors. Atomic coordinates have been deposited in the Protein Data Bank (6BS8).

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

## Acknowledgements

We thank Dr. Kathleen McDonough, Cathleen Schiraldi, Dr. Olga Novikova, and Dr. Valjean Bacot-Davis for valuable exchanges, constructive feedback, and insight. Work in our laboratory is supported by National Institutes of Health grants GM39422, GM44844, 2T32AI055429, and F32GM121000. Use of the Stanford Synchrotron Radiation Light-source, SLAC National Accelerator Laboratory, is supported by the U.S. Department of Energy, Office of Science, Office of Basic Energy Sciences under Contract No. DE-AC02-76SF00515. The SSRL Structural Molecular Biology Program is supported by the DOE Office of Biological and Environmental Research, and by the National Institutes of Health, National Institute of General Medical Sciences (including P41GM103393). The contents of this publication are solely the responsibility of the authors and do not necessarily represent the official views of NIGMS or NIH.

## Author contributions

D.S.K. and M.B. conceived the study; D.S.K., C.W.L., Z.L., H.L. and M.B. designed the experiments; D.S.K. and C.W.L. performed them, and D.S.K., C.W.L. and M.B. analyzed the data; M.R.M. contributed mass spectrometry expertise; D.S.K. provided the over-expression construct and selenomethionine-labeled protein for crystallography; Z.L. and H.L. contributed the crystal structure; N.K.B. contributed the disulfide optimized DnaBi1 model; D.S.K., C.W.L., Z.L., M.R.M., N.K.B., H.L. and M.B. wrote the manuscript.

## Additional information

**Competing interests:** The authors declare no competing interests.

