## [Peer Review File · Nature Communications]

Reviewers' comments:

Reviewer #1 (Remarks to the Author):

Novel findings: control of splicing by hydrogen peroxide in vivo and in vitro, inhibiting splicing of a reporter in the intein's original organism, crystal structure of a class 3 intein.

This paper demonstrates splicing of the two inteins from *M. smegmatis* DnaB. In *E. coli*, DnaBi2 splices so efficiently that no precursor is observed, while ~50% of the DnaBi1 intein failed to splice in both a reporter and the native DnaB precursor. Splicing of the DnaBi1 reporter continued in vitro, reaching ~50% after 24 hours. This suggests that a significant portion of the DnaBi1 precursors may be irreversibly misfolded or otherwise inactivated when expressed in *E. coli*, bringing into question the validity of applying conclusions from *E. coli* experiments to the natural splicing situation in *M. smegmatis*.

Splicing in vitro of the DnaBi1 reporter was inhibited by ROS and RNS reagents resulting in accumulation of TCEP-sensitive precursors with aberrant mobility, while similar treatment of the DnaBi2 reporter had no effect. The oddly migrating precursors had inter- and intra-molecular disulfide bonds involving C48 and C118. C48 was shown to be essential for splicing since no substitution spliced. Splicing of the fast acting DnaBi2 mutant was not responsive to ROS/RNS, but these experiments are hard to interpret because so little DnaBi2 precursor was present at the starting point of treatment.

The authors report the first crystal structure of a class 3 intein, which provides insight into its splicing mechanism and the roles of the WCT motif and C48.

Treatment of *M. smegmatis* containing the DnaBi1 intein in KanR under stringent Kan selection demonstrated reduced viability with peroxide treatment, indicating that splicing in KanR was inhibited in *M. smegmatis* by peroxide treatment. The viability data would be strengthened by quantitating splicing with or without peroxide treatment since in vivo selection only requires a specific amount of KanR, not efficient splicing.

The authors should examine splicing of the native precursor in *M. smegmatis* via western blot or a reporter tag to determine whether slow or partial splicing occurs in the fully native system. This is relevant because (1) inteins often splice with slower kinetics when inserted into reporter proteins even when native extein amino acids are present and (2) inteins that splice slowly are more susceptible to inhibition than inteins that splice rapidly. Otherwise, we are left to wonder if the ROS/RNS effects are only operational when precursors accumulate because of expression in non-native organisms and/or non-native host proteins.

The authors missed the opportunity to directly demonstrate that inteins act as post-translational environmental sensors by examining growth kinetics of *M. smegmatis* with or without hydrogen peroxide treatment. They propose that DnaBi1's ability to sense stress pauses replication until the stress no longer is present. This experiment would unequivocally prove their hypothesis. Peroxide inhibition of growth won't prove that it's the intein, but the absence of inhibition of growth does prove that the DnaBi1 intein isn't acting as a sensor. Without this simple experiment, the paper is just another in a series showing that slowly splicing inteins can be controlled. With this experiment, it becomes a seminal paper directly proving that inteins benefit their organisms by acting as post-translational toggle switches.

The authors should rephrase the first sentence of the abstract to indicate that an increasing number of inteins show sensitivity to environmental conditions, as opposed to implying that all inteins 'are emerging as post-translational environmental sensors'. This sentence is too strong, since only a handful of the hundreds of inteins have been shown to act as post-translation sensors under certain conditions, while most are parasitic mobile elements with HEN domains.

Reviewer #2 (Remarks to the Author):

Inteins are often considered to be mere parasitic gene element without any biological function. If so, why have inteins been consistently maintained in certain locations across different organisms? One answer for the high conservation at specific sites is that inteins are inserted in the active sites of essential proteins, and therefore inteins become difficult to be removed. Another possibility is to provide some advantages to host organisms. Belfort group has been pushing the hypothesis that inteins have some switch function for environmental stresses and conditions. This report by Kelley et al. is yet another hypothesis that DnaB intein in mycobacteria works oxidative stress sensor and also report the crystal structure of Class III intein structure. If the switch function is valid, mycobacteria having inteins with such a switch function should have advantages over mycobacteria without the inteins under stress conditions. The authors demonstrated this with an engineered system using Kanamycin resistance in mycobacteria. Despite the structural basis provided by the crystal structure of the class 3 intein, the experimental data as oxidative stress sensor is not sufficiently convincing (see below also).

In particular, the recent report demonstrated the high fitness cost of having an intein and increased recombination rates (Noar et al. PNAS, 2016, 113), which have provided insights on the persistence of inteins. There are some issues to be addressed in this article.

1. Why did the authors use Kanamycin resistant to monitor DnaB inteins in mycobacteria instead of directly analyzing the fitness of mycobacteria with/without DnaB inteins under the stress conditions?
2. Enzymes relying on Cys for catalysis are intrinsically sensitive to oxidation. This is studied in detail for many examples, e.g. cysteine proteases. Likewise, many enzymes might have other reaction requirements such as a pH optimum for example. However, this by far does not qualify them to be classified as "pH sensor". The word of "oxidative sensor" is an exaggeration of the optimal condition. The sensor or switch should ideally have complete "ON" and "OFF" without backgrounds, particularly for protein engineering purpose.
3. The effect of "splicing regulation" in Figure 3 is so vague, that is hardly can be resolved. If this is caused by the non-natural expression host used (*E. coli*), then the authors should probably perform these kinds of experiments in *Mycobacterium* also to be consistent with their later experiments using the kanamycin selection (Fig. 4).
4. In Figure 3, the authors did not observe any splicing regulation of the BnaB1 intein with 800 μM H_2O_2 . Higher concentrations were needed and also DA, DEA, and AS had stronger effects on splicing inhibition. It is unclear why then in Figure 4 ("splice or die") 250 μM H_2O_2 were sufficient. And why did the authors use H_2O_2 in favor of DA, DEA, and AS which appeared to be better? What happens at higher concentrations and when using the other oxidizing agents?
5. In the discussion (p. 14, l. 375), the authors state "it is unclear if full precursor or partially spliced products like Pi1 retain any DnaB activity". This is a very crucial point and should be addressed in the first place, before any hypothesis on the intein-mediated regulation of DnaB can be drawn at all. The inactivity was assumed for this entire article. If there is no such inhibition, then the entire proposed model might be able to be not justified.
6. It would be interesting to test if sensitivity to oxidation is a general property of class 3 inteins which might be related to the WCT motif uniquely found in this class, or if it applies just to specific inteins like MsmDnaB1. If it is specific to one intein, the switch hypothesis for the persistence of inteins can be a special case rather than a general mechanism.

7. What is the structural role of the conserved Trp residue in the class III motif except for the fact that it is buried in the core? I would expect a more important role as it is essential for the class 3. This could be discussed in detail.

8. In the discussion (p.13, l. 345) the authors propose the MsmDnaB1 intein to be used in the future for biotechnological application because of the natural redox trap. According to the results in Figure 3, the splicing efficiency of the MsmDnaB1 intein is less than 50% and hence does not call for a prominent candidate for such kind of applications. The authors could explain in detail how this feature could be used in any applications by presenting some data as examples.

9. The authors identified two distinct conformations a (36%) and b (64%) for Cys118. Figure 6c could include the electron density map to represent the occupancies. Are there any other secondary conformations in the structure?

10. In the abstract, the authors claim that the class 3 structure can provide a new scaffold for protein engineering. The structure of the class 3 inteins has the identical fold to class 1 intein with the unique sequence. It is unclear how this structure could provide a new scaffold for protein engineering.

11. In Figure 5, the possible Cys48-Cys118 was artificially produced in vitro after purification rather than the isolated protein from in vivo. Thus, it is unclear if the observed disulfide bridge is simply artificially produced and could be irrelevant to biological status.

12. The disulfide bond for Cys48-Cys118 was modeled, but no structural statistics is given for the modeled structure. The validity of the model should be assessed. Did the author try to crystallize the protein without 1 mM DTT to obtain the structure with Cys48-Cys118 bridge?

13. Having multiple conformations does not necessarily indicate "flexible" conformation. It indicates two conformations with the similar energy status, which is separated by a higher energy barrier. The authors need to provide other evidence to support "flexibility" e.g. B-factor, etc., before discussing the flexibility.

Reviewer #3 (Remarks to the Author):

The manuscript "Mycobacterial DnaB helicase intein as oxidative stress sensor" by Kelley et al from Marlene Belfort's lab provides evidence that two inteins that interrupt the DnaB of *M. smegmatis* have differential activity. The second intein, which has a homing endonuclease domain, splices quickly and isn't all that interesting, save for its differential activity to intein one. The first intein is inactivated by a disulfide bond formed under oxidative or nitrosative stress. The authors clearly show that a disulfide bond is formed between the two Cys residues of the intein by MS data. This is interesting and important, but has been shown for other inteins. However, the really important part of the work is that the physiological relevance of the intein switch is demonstrated in the native organism. While other work has shown similar redox controlled effects of inteins expressed in non-native *E. coli*, the discovery that this can occur in *M. smegmatis* is quite important to the authors' premise that the intein can serve as a switch in the native organism, albeit interrupting a different extein. The authors also solve the first structure of a class three intein, and generate clear and informative figures to describe the location and potential roles of key residues in the active site.

This work is on the cutting edge of intein biochemistry, as a potential role for the host organism of a probably limited number of inteins would recast the current understanding of inteins as molecular parasites. The structural work could also permit hypothesis-driven work to understand the activity and flexibility of the class 3 intein active site.

The manuscript is well-written, and the data support the conclusions. It is a very important contribution to the field as written.

The role of Cys48, as the authors note in the discussion, is unclear and could be the subject of future work. Perhaps the disulfide bond is required for proper folding of the intein, or perhaps the Cys48 has a direct chemical role in the mechanism.

The authors might wish to note why the protein band labeled "CTC" in Figure S2 is not the same size for both inteins, if both products should both be GFP with only a four-residue difference in the linker. This is a very minor issue. It would also be interesting for the authors to comment on any statistical significance to the differences in splicing in, for example, Fig. 3b, in which the DnaB1 G-1V intein is said to "accumulate" precursor; it seems more likely that any difference in these lanes is within a margin of error. One could argue that this intein splices before treatment, so it is difficult to draw any conclusion about with the ROS or RNS species would actually affect its splicing.

Reviewer #1:

Novel findings: control of splicing by hydrogen peroxide in vivo and in vitro, inhibiting splicing of a reporter in the intein's original organism, crystal structure of a class 3 intein.

This paper demonstrates splicing of the two inteins from *M. smegmatis* DnaB. In *E. coli*, DnaBi2 splices so efficiently that no precursor is observed, while ~50% of the DnaBi1 intein failed to splice in both a reporter and the native DnaB precursor. Splicing of the DnaBi1 reporter continued in vitro, reaching ~50% after 24 hours. This suggests that a significant portion of the DnaBi1 precursors may be irreversibly misfolded or otherwise inactivated when expressed in *E. coli*, bringing into question the validity of applying conclusions from *E. coli* experiments to the natural splicing situation in *M. smegmatis*.

Several lines of evidence strongly suggest that DnaBi1 folds properly in *E. coli*. First, we have never observed insolubility with purified DnaBi1, even at near mM protein concentrations, and our purification scheme requires the intein to be splicing competent and thus properly folded. Second, DnaBi1 expressed and purified from *E. coli* yielded a crystal structure, which strongly argues against misfolding. Third, the DnaBi1 structure appears as a well-folded protein and similar to reported class 1 intein structures.

Splicing in vitro of the DnaBi1 reporter was inhibited by ROS and RNS reagents resulting in accumulation of TCEP-sensitive precursors with aberrant mobility, while similar treatment of the DnaBi2 reporter had no effect. The oddly migrating precursors had inter- and intra-molecular disulfide bonds involving C48 and C118. C48 was shown to be essential for splicing since no substitution spliced. Splicing of the fast acting DnaBi2 mutant was not responsive to ROS/RNS, but these experiments are hard to interpret because so little DnaBi2 precursor was present at the starting point of treatment.

We agree that MIG DnaBi2 splicing data should be interpreted with caution. We therefore added additional language in **Results**, section “**The two *M. smegmatis* inteins display differential sensitivity to stressors**”, paragraph 2:

“In contrast, MIG DnaBi2 G-1V appeared largely unresponsive to inhibition by these stressors even at a higher magnification or increased contrast, but we cannot exclude the possibility that splicing is still occurring too rapidly for these compounds to show an effect.”

The authors report the first crystal structure of a class 3 intein, which provides insight into its splicing mechanism and the roles of the WCT motif and C48.

We believe that the crystal structure is an invaluable contribution to the intein field, with both mechanistic insight gained into class 3 inteins and a potential for biotechnological applications.

Treatment of *M. smegmatis* containing the DnaBi1 intein in KanR under stringent Kan selection demonstrated reduced viability with peroxide treatment, indicating that splicing in KanR was inhibited in *M. smegmatis* by peroxide treatment. The viability data would be strengthened by

quantitating splicing with or without peroxide treatment since in vivo selection only requires a specific amount of KanR, not efficient splicing.

The advice on quantitation is well taken. Upon further investigation of experimental conditions, we report a much more substantial inhibition of DnaBi1 splicing by H₂O₂ in vivo using our “Splice or Die” reporter. We have included quantitation of this, which shows a relative splicing inhibition of 213 +/- 78-fold following H₂O₂ treatment (Fig. 4d).

The authors should examine splicing of the native precursor in *M. smegmatis* via western blot or a reporter tag to determine whether slow or partial splicing occurs in the fully native system. This is relevant because (1) inteins often splice with slower kinetics when inserted into reporter proteins even when native extein amino acids are present and (2) inteins that splice slowly are more susceptible to inhibition than inteins that splice rapidly. Otherwise, we are left to wonder if the ROS/RNS effects are only operational when precursors accumulate because of expression in non-native organisms and/or non-native host proteins.

To approach this question, we measured by Western blotting if any precursor accumulates in *M. smegmatis* following H₂O₂ treatment. Strikingly, using an Extein1 antibody, we found that the fraction of Pi1 precursor consistently increased relative to ligated exteins (LE) following H₂O₂ treatment by at least 4-fold, to as much as 8-fold. This is an important result and the first report of intein precursor accumulation in a native host in response to stress. We have added these data (Fig. 4e-f) and text to the “**Splicing inhibition of DnaBi1 occurs in *M. smegmatis* under H₂O₂ stress**” section of the **Results**.

The authors missed the opportunity to directly demonstrate that inteins act as post-translational environmental sensors by examining growth kinetics of *M. smegmatis* with or without hydrogen peroxide treatment. They propose that DnaBi1’s ability to sense stress pauses replication until the stress no longer is present. This experiment would unequivocally prove their hypothesis. Peroxide inhibition of growth won’t prove that it’s the intein, but the absence of inhibition of growth does prove that the DnaBi1 intein isn’t acting as a sensor. Without this simple experiment, the paper is just another in a series showing that slowly splicing inteins can be controlled. With this experiment, it becomes a seminal paper directly proving that inteins benefit their organisms by acting as post-translational toggle switches.

We found that peroxide does inhibit *M. smegmatis* growth in liquid culture, and that this inhibition occurs at concentrations of H₂O₂ similar to those that inhibit DnaBi1 splicing *in vitro*, as reported in the paper. Below are the data.

We elected not to include the data in the manuscript because, although suggestive, in the reviewer's words, "Peroxide inhibition of growth won't prove that it's the intein" and believe that the incorporation of the additional *in vivo* data is a more supportive line of evidence of our claims.

The authors should rephrase the first sentence of the abstract to indicate that an increasing number of inteins show sensitivity to environmental conditions, as opposed to implying that all inteins 'are emerging as post-translational environmental sensors'. This sentence is too strong, since only a handful of the hundreds of inteins have been shown to act as post-translation sensors under certain conditions, while most are parasitic mobile elements with HEN domains.

We have adjusted the wording in the **Abstract**:

"Inteins are widespread self-splicing protein elements with an increasing number emerging as post-translational environmental sensors."

Reviewer #2:

Inteins are often considered to be mere parasitic gene element without any biological function. If so, why have inteins been consistently maintained in certain locations across different organisms? One answer for the high conservation at specific sites is that inteins are inserted in the active sites of essential proteins, and therefore inteins become difficult to be removed. Another possibility is to provide some advantages to host organisms. Belfort group has been pushing the hypothesis that inteins have some switch function for environmental stresses and conditions. This report by Kelley et al. is yet another hypothesis that DnaB intein in mycobacteria works oxidative stress sensor and also report the crystal structure of Class III intein structure. If the switch function is valid, mycobacteria having inteins with such a switch function should have advantages over mycobacteria without the inteins under stress conditions. The authors demonstrated this with an engineered system using Kanamycin resistance in mycobacteria. Despite the structural basis provided by the crystal structure of the class 3 intein, the experimental data as oxidative stress sensor is not sufficiently convincing (see below also). In particular, the recent report demonstrated the high fitness cost of having an intein and increased recombination rates (Noar et al. PNAS, 2016, 113), which have provided insights on the persistence of inteins. There are some issues to be addressed in this article.

We find this argument about intein persistence due to increased recombination frequency not to be relevant to our system as the proposed oxidative sensor, DnaBi1, is a “mini intein” lacking a homing endonuclease (HEN) domain. Therefore, it is hard to imagine how the persistence of the DnaBi1 intein could be primarily explained by an increase in recombination frequency. It remains an open question as to why inteins persist in certain genes and organisms, particularly those lacking HEN domains. It is also worth pointing out, as the reviewer notes, that Naor et al. demonstrated an unexpectedly high fitness cost for retaining the intein. In our opinion, this speaks more to the relevance of the intein to the organism than the opposite, as it is retained in spite of the tremendous fitness cost.

1. Why did the authors use Kanamycin resistant to monitor DnaB inteins in mycobacteria instead of directly analyzing the fitness of mycobacteria with/without DnaB inteins under the stress conditions?

Isogenic *M. smegmatis* +/- intein strains do not currently exist, and we have yet been unable to generate these strains. This is an area of high interest and a focus of future work. As a surrogate for these types of *in vivo* experiments, we built the KanR “Splice or Die” reporter to measure splicing inhibition by H₂O₂, showing a greater than 100-fold intein-specific decrease in survival upon H₂O₂ treatment directly in *M. smegmatis*. We have also now demonstrated native Pi1 accumulation under H₂O₂ stress in *M. smegmatis* by Western blotting.

2. Enzymes relying on Cys for catalysis are intrinsically sensitive to oxidation. This is studied in detail for many examples, e.g. cysteine proteases. Likewise, many enzymes might have other reaction requirements such as a pH optimum for example. However, this by far does not qualify them to be classified as “pH sensor”. The word of “oxidative sensor” is an exaggeration of the optimal condition. The sensor or switch should ideally have complete “ON” and “OFF” without backgrounds, particularly for protein engineering purpose.

It seems unrealistic to expect an intein to behave in a strictly On/Off manner, and to our knowledge, no native or engineered intein behaves in this fashion. Further, countless regulatory biological systems do not behave in a strictly binary manner, so we respectfully disagree with the notion that this is a requirement to be of use as a regulatory switch to the host. There are situations where having some level of control over splicing would be beneficial to the host organism, to modulate growth and survival, even if the system was not strictly On/Off. For protein engineering purposes, we view this oxidative control/redox trap as a starting point on which to build, rather than suggesting this is a refined switch.

3. The effect of “splicing regulation” in Figure 3 is so vague, that is hardly can be resolved. If this is caused by the non-natural expression host used (*E. coli*), then the authors should probably perform these kinds of experiments in *Mycobacterium* also to be consistent with their later experiments using the kanamycin selection (Fig. 4).

We have now performed key experiments in *M. smegmatis*. First, we demonstrated enormous DnaBi1 splicing inhibition using our “Splice or Die” genetic reporter, demonstrating a relative splicing inhibition of 213 +/- 78-fold following H₂O₂ treatment (see Figure 4c and d). Second, we are pleased to now include data demonstrating DnaB Pi1 precursor accumulation following H₂O₂ treatment in native *M. smegmatis* (Fig. 4e-f) as detected by Western blotting. Results obtained within *M. smegmatis* using H₂O₂ are entirely consistent with all biochemical data presented, strongly supporting an intein-based regulation of DnaB under H₂O₂ stress.

4. In Figure 3, the authors did not observe any splicing regulation of the BnaB1 intein with 800 μM H₂O₂. Higher concentrations were needed and also DA, DEA, and AS had stronger effects on splicing inhibition. It is unclear why then in Figure 4 (“splice or die”) 250 μM H₂O₂ were sufficient. And why did the authors use H₂O₂ in favor of DA, DEA, and AS which appeared to be better? What happens at higher concentrations and when using the other oxidizing agents?

We attribute variance in effective H₂O₂ concentrations to be due to the difference of an *in vitro* system under specific buffer conditions in solution, or *in vivo* in liquid media, compared to an *in vivo* system on solid growth media. We note it is routine for us to see disparate behavior in the influence of H₂O₂ on *M. smegmatis* growth on plates versus in liquid, with H₂O₂ being significantly more potent on plates. Moreover, we chose to focus on H₂O₂ over other stressors due to it being a highly relevant and well-studied stressor for *Mycobacteria*.

5. In the discussion (p. 14, l. 375), the authors state “it is unclear if full precursor or partially spliced products like Pi1 retain any DnaB activity”. This is a very crucial point and should be addressed in the first place, before any hypothesis on the intein-mediated regulation of DnaB can be drawn at all. The inactivity was assumed for this entire article. If there is no such inhibition, then the entire proposed model might be able to be not justified.

We find it highly unlikely that the Pi1 precursor, or any precursor product, of DnaB can perform complete replication-related activities, such as ATP hydrolysis. However, we are open to the possibility of partial activities, such as DNA binding, as has been shown for intein-containing RadA from *P. horikoshii* (Lennon et al. *Genes Dev*, 2016). Further, work with *P. horikoshii* RadA, which also has an intein inserted within the P-loop, demonstrates that ATPase activity is

abolished prior to the intein splicing (Topilina et al. *NAR*, 2015), strongly suggesting that full activity requires splicing. We have added this point to the **Discussion**, section “**Cysteine toggle for stress sensing and implications for replication**”, paragraph 4:

“It has been shown for another intein, inserted in the P-loop-of *Pyrococcus horikoshii* RadA, that the intein’s presence disrupts ATPase function⁴, and it is reasonable to assume this would be true for P-loop-inserted DnaBi1.”

6. It would be interesting to test if sensitivity to oxidation is a general property of class 3 inteins which might be related to the WCT motif uniquely found in this class, or if it applies just to specific inteins like MsmDnaB1. If it is specific to one intein, the switch hypothesis for the persistence of inteins can be a special case rather than a general mechanism.

We agree with the reviewer that this is an interesting question but is beyond the scope of this manuscript.

7. What is the structural role of the conserved Trp residue in the class III motif except for the fact that it is buried in the core? I would expect a more important role as it is essential for the class 3. This could be discussed in detail.

Work by the Perler group (Tori et al. *J Biol Chem*, 2010; Tori & Perler *J Bacteriol*, 2011) has extensively characterized the biochemistry of several class 3 inteins, including the role of Trp, and our structure supported this role as part of a hydrophobic core.

8. In the discussion (p.13, l. 345) the authors propose the MsmDnaB1 intein to be used in the future for biotechnological application because of the natural redox trap. According to the results in Figure 3, the splicing efficiency of the MsmDnaB1 intein is less than 50% and hence does not call for a prominent candidate for such kind of applications. The authors could explain in detail how this feature could be used in any applications by presenting some data as examples.

Our discussion of biotechnological application is meant merely to emphasize that the new structure, with unique and previously unknown active site features, may serve as a template to engineer new technologies. We have added this point to the **Discussion**, section “**Biochemical and structural insights into an atypical splicing mechanism**”, paragraph 3:

“The structural insight gained from this class 3 intein may provide a novel scaffold in which to engineer and design new splicing-based technologies.”

Further, we respectfully disagree that 100% activity is needed for successful development of an intein-based application, particularly for those applications such as biosensors, where the most important factor is signal to noise rather than efficiency of splicing.

9. The authors identified two distinct conformations a (36%) and b (64%) for Cys118. Figure 6c could include the electron density map to represent the occupancies. Are there any other secondary conformations in the structure?

We have incorporated an electron density map of the two Cys118 conformations into Figure 6, panel d. Additionally, Gln64 in the B chain was found to display a secondary conformation and

this information has been added to the text in the section “**DnaBi1 structure reveals conformational flexibility of the catalytic cysteine**” of the **Results**.

10. In the abstract, the authors claim that the class 3 structure can provide a new scaffold for protein engineering. The structure of the class 3 inteins has the identical fold to class 1 intein with the unique sequence. It is unclear how this structure could provide a new scaffold for protein engineering.

While the class 3 intein shares overall structural similarity to class 1 inteins, the positions of key catalytic residues are distinct and such differences may serve as a basis for hypothesis-driven research, including the exploration of protein engineering and design. We have added additional clarification to the **Discussion**, section “**Biochemical and structural insights into an atypical splicing mechanism**”, paragraph 3:

“...we expect that class 3 inteins hold untapped potential in this arena, particularly since the arrangement of catalytic residues is compacted compared to class 1.”

11. In Figure 5, the possible Cys48-Cys118 was artificially produced in vitro after purification rather than the isolated protein from in vivo. Thus, it is unclear if the observed disulfide bridge is simply artificially produced and could be irrelevant to biological status.

Our demonstration of H₂O₂-based splicing regulation in *M. smegmatis* in the KanR “Splice or Die” system (Fig. 4a-d) indicates that splicing inhibition in response to H₂O₂ is occurring in vivo in a native background. Further, we have added experiments that demonstrate DnaB Pi1 precursor accumulation following H₂O₂ by Western blot (Fig. 4e-f). The biochemical characterization of disulfide bond formation under H₂O₂ conditions demonstrates that this disulfide forms. Taken together, it is reasonable to suggest that this is the mechanism by which splicing inhibition occurs in the native system.

12. The disulfide bond for Cys48-Cys118 was modeled, but no structural statistics is given for the modeled structure. The validity of the model should be assessed. Did the author try to crystallize the protein without 1 mM DTT to obtain the structure with Cys48-Cys118 bridge?

We have included the model statistics into the manuscript (see Supplemental Methods and Table S3). The crystallization of a disulfide-bonded DnaBi1 is a focus of future work and beyond the scope of this manuscript.

13. Having multiple conformations does not necessarily indicate “flexible” conformation. It indicates two conformations with the similar energy status, which is separated by a higher energy barrier. The authors need to provide other evidence to support “flexibility” e.g. B-factor, etc., before discussing the flexibility.

We have included the B-factor data to support our claim of flexibility (Table S1). B factors for Cys118 in the A, B, and C chains are as listed:

Chain A, Cys118: 51.17 Å²

Chain B, Cys118a: 29.66 Å²

Chain B, Cys118b: 41.34 Å²

Chain C, Cys118: 54.90 Å²

Reviewer #3:

The manuscript “Mycobacterial DnaB helicase intein as oxidative stress sensor” by Kelley et al from Marlene Belfort’s lab provides evidence that two inteins that interrupt the DnaB of *M. smegmatis* have differential activity. The second intein, which has a homing endonuclease domain, splices quickly and isn’t all that interesting, save for its differential activity to intein one. The first intein is inactivated by a disulfide bond formed under oxidative or nitrosative stress. The authors clearly show that a disulfide bond is formed between the two Cys residues of the intein by MS data. This is interesting and important, but has been shown for other inteins. However, the really important part of the work is that the physiological relevance of the intein switch is demonstrated in the native organism. While other work has shown similar redox controlled effects of inteins expressed in non-native *E. coli*, the discovery that this can occur in *M. smegmatis* is quite important to the authors’ premise that the intein can serve as a switch in the native organism, albeit interrupting a different extein. The authors also solve the first structure of a class three intein, and generate clear and informative figures to describe the location and potential roles of key residues in the active site.

This work is on the cutting edge of intein biochemistry, as a potential role for the host organism of a probably limited number of inteins would recast the current understanding of inteins as molecular parasites. The structural work could also permit hypothesis-driven work to understand the activity and flexibility of the class 3 intein active site.

The manuscript is well-written, and the data support the conclusions. It is a very important contribution to the field as written.

We appreciate the reviewer’s enthusiasm for the work.

The role of Cys48, as the authors note in the discussion, is unclear and could be the subject of future work. Perhaps the disulfide bond is required for proper folding of the intein, or perhaps the Cys48 has a direct chemical role in the mechanism.

We agree and are thank the reviewer for their thoughts.

The authors might wish to note why the protein band labeled “CTC” in Figure S2 is not the same size for both inteins, if both products should both be GFP with only a four-residue difference in the linker. This is a very minor issue.

The samples were run under non-reducing conditions with the loading dye lacking β mercaptoethanol. The two constructs differ by 7 to 10 amino acids and we attribute the differences in the CTC products to charge differences in the amino acid composition. This is something we’ve observed in other MIG constructs and have adjusted the text in the Figure S2 legend to reflect this:

“Samples are run under non-reducing conditions with loading dye lacking β -mercaptoethanol. The migration difference between the C-terminal cleavage (CTC) products is attributable to distinct C-extein residue composition between *Mle* DnaBi and *Mtu* DnaBi, resulting in charge variances which can cause spurious migration.”

This has been previously observed by Topilina et al. *PNAS*, 2015.

It would also be interesting for the authors to comment on any statistical significance to the differences in splicing in, for example, Fig. 3b, in which the DnaB1 G-1V intein is said to “accumulate” precursor; it seems more likely that any difference in these lanes is within a margin of error. One could argue that this intein splices before treatment, so it is difficult to draw any conclusion about with the ROS or RNS species would actually affect its splicing.

We agree that MIG DnaBi2 splicing data should be interpreted with caution. We therefore added additional language in **Results**, section “**The two *M. smegmatis* inteins display differential sensitivity to stressors**”, paragraph 2:

“In contrast, MIG DnaBi2 G-1V appeared largely unresponsive to inhibition by these stressors even at a higher magnification or increased contrast, but we cannot exclude the possibility that splicing is still occurring too rapidly for these compounds to show an effect.”

REVIEWERS' COMMENTS:

Reviewer #1 (Remarks to the Author):

The authors have adequately responded to my initial critique.

However, I still believe that the growth curve in the rebuttal letter should be included in the supplemental figures or at least mention the experiment in the text with the caveat that it doesn't prove that splicing inhibition by ROS is causing growth inhibition. We all agree that the peroxide treatment is probably affecting multiple enzymes/pathways, but an absence of growth inhibition would have been catastrophic for the author's conclusions. It is the first thought of a reader that the authors haven't reported this key result, so it must not have worked.....

I recommend acceptance after the following minor changes.

1) Abstract: It would be more acceptable to rephrase sentence 1 to reflect the controversy in the field by changing to:

"emerging as 'potential' post-translational"

2) Abstract and discussion: minor point for the authors to consider: there are enough examples of multi-intein proteins to argue that this isn't 'unusual', but instead is 'less common'. Unusual in my mind suggests novel whereas less common suggests that it occurs often enough not to be novel.

3) Line 48: the sentence as written is ambiguous to non-intein researchers. Please clarify, for example:

"Even inteins found in the same protein can differ in 'position and sequence' between host bacteria"

4) Line 64: the introduction is incomplete without adding that previous thoughts on intein maintenance focused on efficient insertion of selfish intein genes and difficulties in deleting insertions at critical positions in a protein.

5) Figure 6: I would have preferred to see the panel with Cys118 distances to Asn139 and Ala1 in the main paper and not the supplemental information. It clearly supports the structure as an active conformation.

6) Line 312: Consider rephrasing to clarify "Further, we present in vivo support of DnaBi1 splicing inhibition through detection of Pi1 precursor accumulation in *M. smegmatis* following H₂O₂ treatment by Western blotting" as "Further, we present in vivo support of DnaBi1 splicing inhibition through detection 'by Western blotting' of Pi1 precursor accumulation in *M. smegmatis* following H₂O₂ treatment."

7) Line 314: The way this is written suggests to me that the disulfide bond is required for initiating splicing instead of locking the intein. Please consider rephrasing.

8) Line 337: I strongly disagree with the discussion point that mini-inteins 'must become useful'. As many groups have stated, it is very difficult to remove an intein in an enzyme active site without inactivating the host protein. This alone can be sufficient to keep the intein, without a mandatory requirement for further usefulness. It is misleading to make this usefulness statement without including the alternative possibility that it is just hard to remove inteins in active sites.

REVIEWERS' COMMENTS:

Reviewer #1 (Remarks to the Author):

The authors have adequately responded to my initial critique.

However, I still believe that the growth curve in the rebuttal letter should be included in the supplemental figures or at least mention the experiment in the text with the caveat that it doesn't prove that splicing inhibition by ROS is causing growth inhibition. We all agree that the peroxide treatment is probably affecting multiple enzymes/pathways, but an absence of growth inhibition would have been catastrophic for the author's conclusions. It is the first thought of a reader that the authors haven't reported this key result, so it must not have worked.....

We have incorporated the growth curve into the manuscript as Supplementary Figure 3 and added additional text regarding the experiment.

I recommend acceptance after the following minor changes.

1) Abstract: It would be more acceptable to rephrase sentence 1 to reflect the controversy in the field by changing to:

"emerging as 'potential' post-translational"

We have added the requested language to the abstract.

2) Abstract and discussion: minor point for the authors to consider: there are enough examples of multi-intein proteins to argue that this isn't 'unusual', but instead is 'less common'. Unusual in my mind suggests novel whereas less common suggests that it occurs often enough not to be novel.

We have adjusted the text in the abstract and discussion accordingly.

3) Line 48: the sentence as written is ambiguous to non-intein researchers. Please clarify, for example: "Even inteins found in the same protein can differ in 'position and sequence' between host bacteria"

We have clarified the text as recommended.

4) Line 64: the introduction is incomplete without adding that previous thoughts on intein maintenance focused on efficient insertion of selfish intein genes and difficulties in deleting insertions at critical positions in a protein.

We appreciate the reviewer concerns and have incorporated this point into the introduction.

Introduction, paragraph 4: "Intein maintenance has generally been attributed to the difficulties associated with precisely removing the intein without fatally disrupting the host protein and the stability of the insertion site sequences, as inteins are often found in highly conserved regions of proteins."

5) Figure 6: I would have preferred to see the panel with Cys118 distances to Asn139 and Ala1 in the main paper and not the supplemental information. It clearly supports the structure as an active

conformation.

We have moved the panel showing the distances in the DnaBi1 active center to main text Figure 6.

6) Line 312: Consider rephrasing to clarify “Further, we present in vivo support of DnaBi1 splicing inhibition through detection of Pi1 precursor accumulation in *M. smegmatis* following H₂O₂ treatment by Western blotting” as “Further, we present in vivo support of DnaBi1 splicing inhibition through detection 'by Western blotting' of Pi1 precursor accumulation in *M. smegmatis* following H₂O₂ treatment.”

We have made the suggested text change.

7) Line 314: The way this is written suggests to me that the disulfide bond is required for initiating splicing instead of locking the intein. Please consider rephrasing.

We have added additional clarifying language.

Discussion, Paragraph 1: “Biochemical and structural characterization of DnaBi1 establish that an unusual cysteine is required for splicing and, **under oxidative stress**, forms an intramolecular disulfide bond with the initiating cysteine nucleophile.”

8) Line 337: I strongly disagree with the discussion point that mini-inteins ‘must become useful’. As many groups have stated, it is very difficult to remove an intein in an enzyme active site without inactivating the host protein. This alone can be sufficient to keep the intein, without a mandatory requirement for further usefulness. It is misleading to make this usefulness statement without including the alternative possibility that it is just hard to remove inteins in active sites.

We have adjusted the text to better reflect the idea of intein maintenance through non-adaptive means.

Discussion, paragraph 3: “...whereas inteins that have lost the HEN domain and mobile properties **must rely on alternative approaches to maintenance. The removal of the intein sequence without inactivating the host protein is difficult but may suffice for long-term intein survival. Alternatively, the intein may become adapted to the host and thus be maintained by serving a function.**”